# Symmetry in language statistics shapes the geometry of model representations

Dhruva Karkada [1] [*]   Daniel J. Korchinski [2]   Andres Nava [3]   Matthieu Wyart [2] [3]   Yasaman Bahri [4]

## Abstract

The internal representations learned by language models consistently exhibit striking geometric structure: calendar months organize into a circle, historical years form a smooth one-dimensional manifold, and cities' latitudes and longitudes can be decoded using a linear probe. To explain this neural code, we first show that language statistics exhibit translation symmetry (for example, the frequency with which any two months co-occur in text depends only on the time interval between them). We prove that this symmetry governs these geometric structures in high-dimensional word embedding models, and we analytically derive the manifold geometry of word representations. These predictions empirically match large text embedding models and large language models. Moreover, the representational geometry persists at moderate embedding dimension even when the relevant statistics are perturbed (e.g., by removing all sentences in which two months co-occur). We prove that this robustness emerges naturally when the co-occurrence statistics are controlled by an underlying latent variable. Our results indicate that these representational manifolds originate in the statistical symmetries of natural language.

## 1. Introduction

The semantic information encoded in the trained parameters of an LLM is reflected in the vector geometry of its internal representations. A growing body of empirical work has documented a number of intriguing geometric structures in this neural code, including prisms (Mikolov et al., 2013; Park et al., 2024; Merullo et al., 2024), loops (Engels et al., 2025; Modell et al., 2025), and continuous 1D manifolds (Gurnee et al., 2025). Notably, these structures emerge consistently across a range of model architectures and tasks. Despite this apparent universality, we lack an organizing principle that explains why these patterns arise.

---
[*]Work done in part during an internship at Google DeepMind. [1]UC Berkeley [2]EPFL [3]Johns Hopkins [4]Google DeepMind. Correspondence: dkarkada@berkeley.edu, daniel.korchinski@epfl.ch, anava1@jh.edu, mwyart1@jh.edu, yasamanbahri@gmail.com.

*Proceedings of the 43rd International Conference on Machine Learning*, Seoul, South Korea. PMLR 306, 2026. Copyright 2026.

In this work, we provide such an organizing principle: that representational geometry reflects pairwise co-occurrence statistics between words. In particular, we give evidence that *symmetry governs these statistics and drives the formation of representational manifolds*. This hypothesis underpins our theory and explains the following phenomena:

- **Circles in representation space for cyclical concepts.** Engels et al. (2025); Modell et al. (2025) find that LLMs encode concepts such as days of the week, months of the year, or hues on the color wheel with circular geometry. Gromov (2023) and related work suggest that circular representations may enable LLMs to efficiently compute modular addition tasks ("Five months after November is:").

- **"Rippled" 1D manifolds for continuous sequences.** Engels et al. (2025); Gurnee et al. (2025) find that LLM representations corresponding to continuums (e.g., years of history or number lines) are organized in compact 1D manifolds with so-called "ripples," or extrinsic curvature. The LLM exploits this manifold geometry to compute increments and decrements.

- **Linear decoding of spatiotemporal coordinates.** Gurnee & Tegmark (2024) find that LLM representations of geographical locations and historical events *linearly* encode their spatial and temporal coordinates, respectively. Thus, LLMs may straightforwardly compute maps and timelines by linear transformations.

To test this hypothesis, we first examine word embedding models, where representation learning is driven exclusively by pairwise co-occurrence statistics. Surprisingly, we find that each of these geometries arises in this simple setting. We develop a mathematical theory that unifies and explains these seemingly disparate observations, deriving the representation geometries from translation symmetry in the pairwise co-occurrence statistics of words. For example, when the co-occurrence probability of two events or places depends only on their temporal or spatial distance, these models spontaneously learn Fourier representations, i.e., the embedding vectors vary sinusoidally along each principal component. We analytically predict both the amplitude and the frequency of each component, showing that loops and circles in the manifold geometry correspond to the long-wavelength modes, while ripples are the higher harmonics.

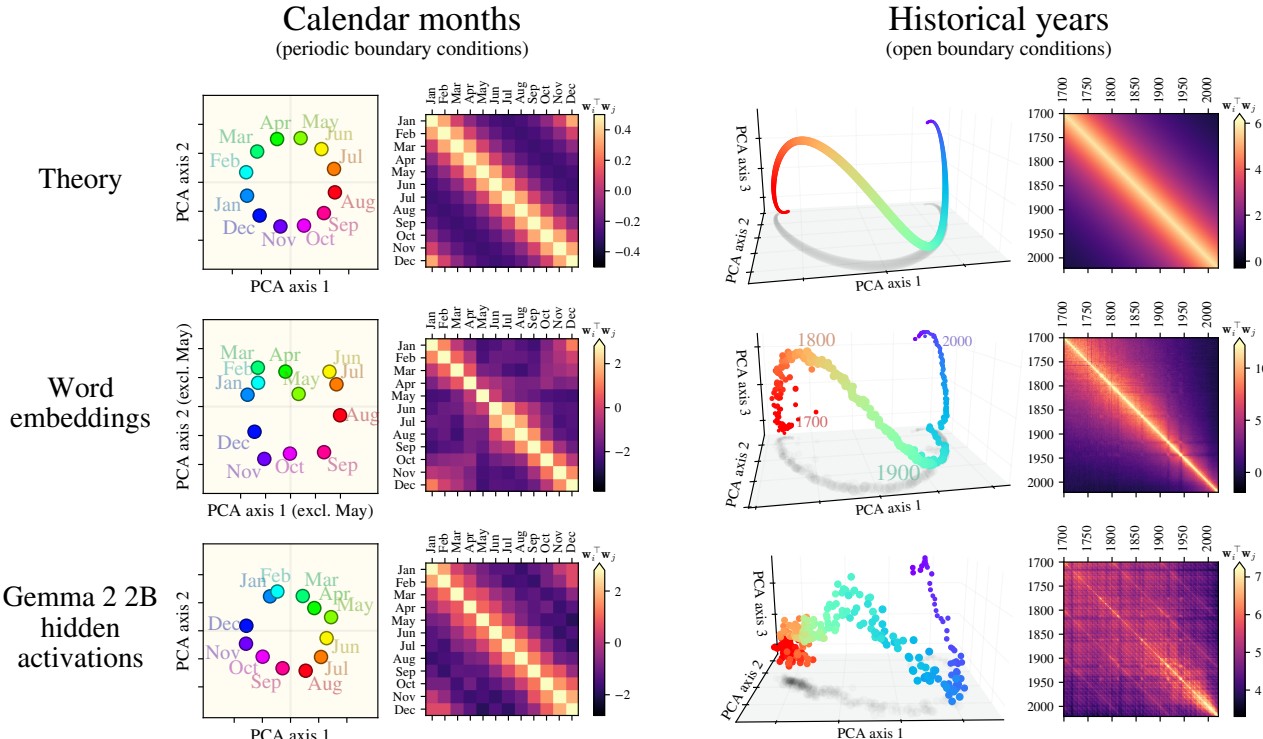

*Figure 1.* **Cyclic representation manifolds arise from translation symmetry in co-occurrence statistics.** We visualize the top principal components of the word representations for calendar time and historical time, and we show the representation vectors' Gram matrix. In the top row, we predict both embedding geometry and their Gram matrix using the parametric curves analytically derived in Corollary 2 and Proposition 3. We empirically compare these predictions to both word embeddings trained on Wikipedia (middle row) and to LLM internal representations (bottom row). The excellent agreement provides evidence that translation symmetry in co-occurrence statistics drives the formation of representational manifolds in neural networks. See Appendix A for experimental details.

Importantly, even if the co-occurrence statistics of a given set of words is perturbed (e.g., removing all sentences where two months co-occur), their representational geometry is preserved at intermediate embedding dimension. One of our central results is to explain this surprising robustness: it emerges because the co-occurrence of many words in the vocabulary are affected by the corresponding latent variable (e.g., the season/time of year), leading to large eigenvalues in the pairwise statistics of words whose eigenvectors are insensitive to noise or perturbation.

In summary, our concrete contributions are:

- Empirical observations of representation manifolds (Figure 1) and linear coordinate decoding (Figure 2) in shallow word embedding models (e.g., word2vec).

- Empirical evidence that words embodying a continuous latent concept have pairwise co-occurrence statistics that exhibit translation symmetry (Figures 5 and 6).

- Analytical expressions that predict the representational geometry directly from these translation symmetric statistics (Corollary 2 and Proposition 3).

- Empirical validation that the predicted representational structure is present, stable, and interpretable in both word embedding models and deep transformer-based models (Figures 1 and 3).

- Theoretical analysis that predicts how the error in the linear coordinate decoding task scales with the probe dimension (Proposition 4) and empirical validation in Figure 2.

- Empirical evidence that representation manifolds are robust to significant perturbations in their co-occurrence statistics even at intermediate embedding dimension (Figure 4).

- A continuous latent variable model that explains this surprising observation, showing that the emergence of representational geometric structure is a collective phenomenon involving many words (Section 4).

Our work demonstrates how structure in the data—namely, symmetry in low-order token correlations—shapes learned representations in neural networks across a variety of model architectures and learning tasks.

## 2. Preliminaries and Related Work

We use capital boldface to denote matrices and lowercase boldface for vectors. Parenthesized subscripts denote tensor elements (e.g., $\boldsymbol{A}_{(ij)}$ is a scalar).

The training corpus is a long sequence of tokens (or words) drawn from a vocabulary of size $V$. We reserve $i$ and $j$ for vocabulary indices. For any two tokens $i$ and $j$, we use $P_{ij}$ to denote their co-occurrence probability, which represents how frequently they appear within a shared context window in natural text. Similarly, $P_i := \sum_j P_{ij}$ is the unigram probability of word $i$. See Section A.2 for formal definitions.

### 2.1. Predicting word embedding geometry

Pairwise co-occurrences are arguably the simplest nontrivial statistic of natural language data. Well-known methods such as `word2vec`, GloVe, and Latent Semantic Analysis (LSA) (Mikolov et al., 2013; Pennington et al., 2014; Landauer & Dumais, 1997) learn exclusively from this statistic. In particular, word embedding models learn an embedding matrix $\boldsymbol{W} \in \mathbb{R}^{V \times d}$ whose $i^{\text{th}}$ row $\boldsymbol{w}_i$ is the $d$-dimensional embedding vector for word $i$. In recent work, Karkada et al. (2025) showed that such models learn to represent the top eigenmodes of a *normalized co-occurrence matrix*, approximately equal to the pointwise mutual information (PMI) matrix:

$$\boldsymbol{M}^{\star}_{(ij)} := \frac{P_{ij} - P_i P_j}{\frac{1}{2}(P_{ij} + P_i P_j)} \approx \log\left(\frac{P_{ij}}{P_i P_j}\right). \quad (1)$$

In terms of the eigendecomposition $\boldsymbol{M}^{\star} = \boldsymbol{\Phi}\boldsymbol{\Lambda}^{\star}\boldsymbol{\Phi}^{\top}$, the learned embeddings are

$$\boldsymbol{W}_{(i\mu)} = \boldsymbol{\Phi}_{(i\mu)}\sqrt{|\boldsymbol{\Lambda}^{\star}_{(\mu\mu)}|}. \quad (2)$$

This key result states that the $\mu^{\text{th}}$ principal component of the embeddings encodes the $\mu^{\text{th}}$-largest eigenmode of $\boldsymbol{M}^{\star}$, i.e., $\boldsymbol{\Phi}_{(\cdot\mu)}$. This insight provides an analytical link between co-occurrence statistics and learned embedding geometry, and thus it underlies our theoretical analysis. In Appendix B we review this result, including a complete discussion of $\boldsymbol{M}^{\star}$, its interpretation, and its relation to the PMI matrix.

Although this result gives exact expressions for the learned embeddings, it requires numerically diagonalizing $\boldsymbol{M}^{\star}$ in general. In certain cases, however, $\boldsymbol{M}^{\star}$ may be dominated by a simple structure that may be *analytically* diagonalized. A recent example of this is the work of Korchinski et al. (2025), which shows that the parallelogram-like geometry underlying linear analogies is a consequence of Kronecker structure in $\boldsymbol{M}^{\star}$. In particular, when the co-occurrence statistics are driven by weakly correlated discrete latent variables, analogy completion by vector addition follows naturally.

Our approach mirrors this logic. We show that the curved geometry underlying the observed "feature manifolds" is a consequence of *translation symmetry* in $\boldsymbol{M}^{\star}$. In particular, when the corpus co-occurrence statistics[1] are driven by a *continuous* latent variable, both the linear decoding property and robustness to noise follow naturally.

To fully describe the mutual word embedding geometry for some subset $\mathcal{S}$ of the vocabulary, it is sufficient to specify their PCA coordinates. Let $\boldsymbol{W}_{\mathcal{S}} := \boldsymbol{P}\boldsymbol{W}_{([\mathcal{S}],\cdot)} \in \mathbb{R}^{|\mathcal{S}| \times d}$ be the centered embeddings for $\mathcal{S}$, where $\boldsymbol{P} = \boldsymbol{I} - |\mathcal{S}|^{-1}\mathbf{1}\mathbf{1}^{\top}$ is the mean-centering matrix. Let its compact SVD be $\boldsymbol{W}_{\mathcal{S}} = \boldsymbol{\Phi}\boldsymbol{\Lambda}^{1/2}\boldsymbol{U}^{\top}$. Then to describe its PCA geometry, it suffices to predict the elements of $\overline{\boldsymbol{W}}_{\mathcal{S}} := \boldsymbol{W}_{\mathcal{S}}\boldsymbol{U} = \boldsymbol{\Phi}\boldsymbol{\Lambda}^{1/2}$. The projected embeddings $\overline{\boldsymbol{W}}_{\mathcal{S}} \in \mathbb{R}^{|\mathcal{S}| \times \min(|\mathcal{S}|,d)}$ retain all information contained in the subspace spanned by $\boldsymbol{W}_{\mathcal{S}}$, thus fully recovering the Gram matrix: $\boldsymbol{W}_{\mathcal{S}}\boldsymbol{W}_{\mathcal{S}}^{\top} = \overline{\boldsymbol{W}}_{\mathcal{S}}\overline{\boldsymbol{W}}_{\mathcal{S}}^{\top}$.

Note that if we approximate $\boldsymbol{M}^{\star}$ as positive semi-definite and if the embedding dimension satisfies $d \geq \operatorname{rank}\boldsymbol{M}^{\star}$, then the uncentered Gram matrix must exactly recover the relevant submatrix of $\boldsymbol{M}^{\star}$, denoted $\boldsymbol{M}^{\star}_{\mathcal{S}}$. Therefore $\overline{\boldsymbol{W}}_{\mathcal{S}}\overline{\boldsymbol{W}}_{\mathcal{S}}^{\top} = \boldsymbol{P}\boldsymbol{M}^{\star}_{\mathcal{S}}\boldsymbol{P}^{\top}$. The theoretical results in Section 3 follow from this key fact.

### 2.2. Related work

Engels et al. (2025); Gurnee & Tegmark (2024); Gurnee et al. (2025) identify examples of highly structured geometry for the representations of spatial, temporal, and numerical concepts within language models. These are motivating examples for our work; we unify them with a single principle rooted in symmetry and we study them analytically in the setting of word embedding models.

Cagnetta & Wyart (2024); Favero et al. (2025); Rende et al. (2024) show that LLMs learn low-order token correlations—e.g., the pairwise statistics captured by word embedding models—before higher-order ones. This suggests that large language models learn contextualized representations and computational circuits atop coarse low-order statistics of natural language. In this view, it is natural that symmetry in pairwise co-occurrence statistics remains a primary factor in shaping model representations.

Saxe et al. (2019) show that when a linear model is trained on synthetic data derived from a periodic lattice, its hidden representation geometry is circular. However, this work does not consider the self-supervised natural language setting where the corpus statistics are key. In LLMs, Park et al. (2025a) observe representational manifolds shaped *in-context* when the prompt is a random walk on a lattice;

---

[1]We use the shorthand "co-occurrence statistics" to refer to the composite quantity $\boldsymbol{M}^{\star}$ rather than the raw probabilities $P_{ij}$. This is because the spectral structure of $\boldsymbol{M}^{\star}$ is dominated by $P_{ij}$ rather than the unigram component $P_i P_j$.

Noroozizadeh et al. (2025) observe similar geometries when training transformers on graphs, showing that the geometry captures top eigenvectors of the underlying graph adjacency matrix. In natural data, Prieto et al. (2026) observe cyclic correlations between months in Wikipedia and empirically find that nonlinear autoencoders learn circular representations by approximately performing PCA.

We go beyond these works by 1) providing a mechanism that explicitly connects symmetry in the data to representational geometry; 2) demonstrating and theoretically explaining the phenomenon in word embedding models, arguably the simplest language models; 3) giving a detailed and unified mathematical description of the representation manifolds by handling open boundary conditions and 2-dimensional lattices; and 4) importantly, identifying and characterizing the collective nature of the phenomenon.

## 3. Co-occurrence model with symmetry

We aim to describe the representational geometry of words and phrases that share an underlying continuous concept such as the time of year, historical time, or geography. A given semantic continuum is represented by a subset $\mathcal{S}$ of the vocabulary (e.g., the 12 months of the year or the 50 US states). We use $D$ to denote the dimension of the underlying semantic continuum, and we say a semantic continuum has "periodic boundary conditions" (BC) if it wraps around, and "open BC" otherwise. As examples, calendar time and the color wheel have $D = 1$ with periodic BC; historical time and number lines have $D = 1$ with open BC; and geography of some sub-region of the globe has $D = 2$ with open BC.

We endow each such concept continuum with a coordinate system, defining $\boldsymbol{x}_i \in \mathbb{R}^D$ as the latent coordinate of word $i \in \mathcal{S}$. For example, $x_{\text{april}} = 4$, and $\boldsymbol{x}_{\text{paris}} = [48.86, 2.35]$ using geographic coordinates. We then define $\text{dist}(\boldsymbol{x}, \boldsymbol{x}')$ as the minimum Euclidean distance between $\boldsymbol{x}$ and $\boldsymbol{x}'$, giving a single notion of distance for both periodic and open BCs.

Intuitively, we expect that in the absence of other strongly discriminating features, no particular location on a semantic continuum is distinguished over any other. If this semantic symmetry holds, its signatures should appear in the corpus statistics; for example, the co-occurrence probability of two months would depend only on the time interval between them, with no dependence on absolute calendar position. We model this *translation symmetry* in the co-occurrence statistics as follows:

**Assumption 3.1. Translation symmetry with kernel** $C(\cdot)$. Let $\tilde{C}(\cdot)$ be some well-behaved function. For any two words $i, j \in \mathcal{S}$, assume $P_{ij} = P_i P_j \tilde{C}(\text{dist}(\boldsymbol{x}_i, \boldsymbol{x}_j))$. It follows that the co-occurrence matrix $\boldsymbol{M}^\star$ (defined in Equation (1)) inherits the translation symmetry:

$$\boldsymbol{M}^\star_{(ij)} = f(\tilde{C}(\text{dist}(\boldsymbol{x}_i, \boldsymbol{x}_j))) =: C(\text{dist}(\boldsymbol{x}_i, \boldsymbol{x}_j)), \quad (3)$$

where $f(y) = 2(y - 1)/(y + 1)$. Thus, the *co-occurrence convolution kernel* $C$ is simply a reparameterization of $\tilde{C}$. Furthermore, assume that $\boldsymbol{M}^\star$ is positive semi-definite.

Assumption 3.1 posits that the co-occurrence probability of any two words in $\mathcal{S}$ depends only on their distance on the semantic continuum. This translation symmetry is a strong constraint on the co-occurrence statistics; nonetheless, we show in Figures 5 and 6 that it accurately describes real data. We assume $\boldsymbol{M}^\star$ is positive semi-definite simply for ease of presentation; in Appendix B.3 we relax this assumption and show that our predictions still hold.[2]

The co-occurrence kernel $C(\cdot)$ captures the functional form of the pairwise correlations. It plays an important role in determining the embedding geometry; in Proposition 1, we show that its Fourier spectrum is connected to the PCA spectrum of the representations.

In the theoretical treatment, we always choose a coordinate system satisfying $\boldsymbol{x}_i \in [-1, 1]^D$. We are afforded this freedom since our theory depends only on the structure of $\boldsymbol{M}^\star$; under Assumption 3.1, rigid transformations of the latent coordinates leave $\boldsymbol{M}^\star$ invariant, and isotropic rescaling induces only a scalar reparameterization of $C$.

Theoretical analysis is simplest when the words of interest are uniformly spaced on the latent continuum (e.g., the calendar months or historical years). In this case, the words occupy sites on a latent semantic *lattice*.

We begin with a general result: Proposition 1 predicts the geometry of learned word embeddings for words on a semantic lattice with arbitrary dimension $D$, arbitrary kernel $C(\cdot)$, and periodic BCs. The theory directly specifies the vector components of the embeddings in their PCA basis.

**Proposition 1** (Fourier embedding geometry (simplified)). *Let $\mathcal{S}$ correspond to a periodic semantic lattice. Assume Assumption 3.1 holds on $\mathcal{S}$ with kernel $C(\cdot)$. Assume the embedding dimension $d \geq \text{rank}(\boldsymbol{M}^\star)$. Define the set of allowed wavevectors $\{\boldsymbol{k}_\mu\}_{\mu=1}^{|\mathcal{S}|}$, each satisfying $\boldsymbol{k} = \pi \boldsymbol{n}$ for some $\boldsymbol{n} \in \mathbb{Z}^D$. Define $\tilde{m}(\boldsymbol{k})$ to be the discrete Fourier transform of $m(\boldsymbol{x}) := C(\text{dist}(\boldsymbol{x}, \boldsymbol{0}))$. Define $\lambda_\mu := \tilde{m}(\boldsymbol{k}_\mu)$. Then the PCA-projected embeddings for the words $i \in \mathcal{S}$ have Fourier geometry:*

$$\overline{\boldsymbol{w}}_i = \sqrt{2/|\mathcal{S}|}\Big[\sqrt{\lambda_1}\cos(\boldsymbol{k}_1^\top \boldsymbol{x}_i), \ \sqrt{\lambda_1}\sin(\boldsymbol{k}_1^\top \boldsymbol{x}_i),$$
$$\sqrt{\lambda_2}\cos(\boldsymbol{k}_2^\top \boldsymbol{x}_i), \ \sqrt{\lambda_2}\sin(\boldsymbol{k}_2^\top \boldsymbol{x}_i), \quad (4)$$
$$\sqrt{\lambda_3}\cos(\boldsymbol{k}_3^\top \boldsymbol{x}_i), \ \dots \Big].$$

---

[2]In Section 4, we show that the continuum coordinates can be seen as a latent variable that underlies and modulates the co-occurrence statistics. In the latent variable model, the symmetry condition follows by construction rather than by direct assumption.

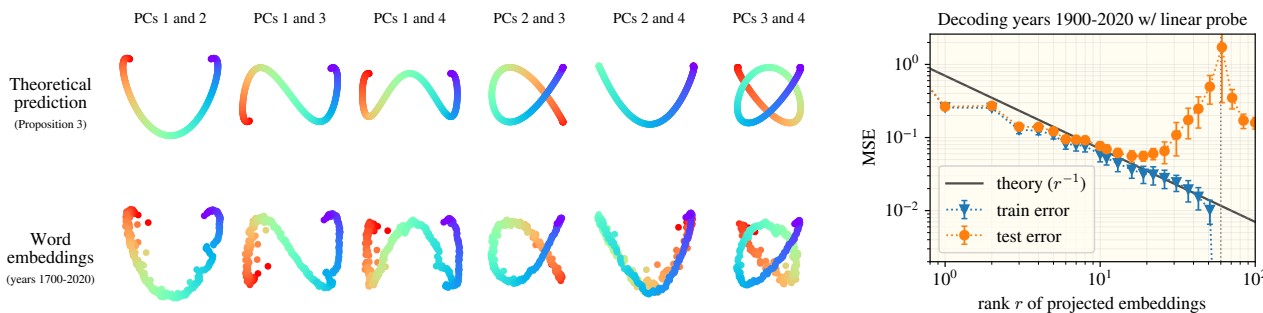

*Figure 2.* **Lattice translation symmetry implies Lissajous curves in embedding space and linear coordinate decoding.** We empirically validate two novel predictions of our theory. **(Left.)** When projected on any two principal components, word embeddings (e.g., for historical years) display Lissajous curves whose amplitudes, phases, and frequencies are analytically predicted by Proposition 3. **(Right.)** Linear probes can decode the numerical year from the sinusoidal embedding modes. Each included higher-frequency mode increases the temporal resolution; the error decay rate is predicted by Proposition 4. At the interpolation threshold $r = n_{\text{train}}$, training error quickly vanishes and the test error exhibits the expected double descent peak.

Put simply, when the latent semantic lattice has periodic BC, each principal component of the learned representations encodes a sinusoidal function of the latent coordinate. Since $m$ is an even function $m(\boldsymbol{x}) = m(-\boldsymbol{x})$, the Fourier modes come in degenerate sin and cos pairs. In each degenerate subspace, the amplitude is given by a Fourier coefficient of the co-occurrence kernel. See Appendix E.2 for the precise statement and proof; the proof follows from the fact that circulant matrices are diagonalized by the discrete Fourier transform.

We now focus on $D = 1$ semantic continuums, which are both simple and prevalent in natural language. Furthermore, in concepts such as calendar time and historical time, we empirically find that the true co-occurrence statistics are well-described by an exponential co-occurrence kernel (i.e., $C(\Delta x; \sigma) \sim \exp(-\Delta x/\sigma)$; see Figures 5 and 6). We will exploit this additional structure to *analytically predict the full parametric curve* for the learned 1-dimensional feature manifolds.

**Corollary 2** (Periodized exponential kernel (simplified)). *Let $\mathcal{S}$ correspond to a $D = 1$ semantic lattice with periodic BC. Let Assumption 3.1 hold on $\mathcal{S}$ with exponential kernel $C(\Delta x) = \sum_{n \in \mathbb{Z}} \exp(-|\Delta x + 2n|/\sigma)$, where $\sigma$ is a free parameter. Assume the embedding dimension is large, i.e., $d \geq \text{rank}(\boldsymbol{M}^\star)$. Then the PCA-projected embeddings for the words $i \in \mathcal{S}$ are given by*

$$\overline{\boldsymbol{w}}_i = \sqrt{2/|\mathcal{S}|}\Big[a_1 \cos(k_1 x_i),\ a_1 \sin(k_1 x_i),$$
$$a_2 \cos(k_2 x_i),\ a_2 \sin(k_2 x_i), \qquad (5)$$
$$a_3 \cos(k_3 x_i),\ \dots\Big]$$

*where*

$$k_n = \pi n \quad and \quad \lim_{|\mathcal{S}| \to \infty} a_n = \sqrt{\frac{2\sigma}{1 + \sigma^2 k_n^2}}. \quad (6)$$

For calendar words, Corollary 2 predicts that the feature manifold is a closed loop whose components oscillate with calendar time with integer frequencies. This prediction is empirically validated in Figures 1 and 9. Appendix E.3 has the precise statement of our general result and its proof, which follows directly from Proposition 1 on the periodized exponential kernel.[3] We now turn to $D = 1$ lattices with *open BC*, where Proposition 1 does not apply.

**Proposition 3** (Exponential kernel, open BC (simplified)). *Let $\mathcal{S}$ correspond to a $D = 1$ semantic lattice with open BC. Let Assumption 3.1 hold on $\mathcal{S}$ with exponential kernel $C(\Delta x) = \exp(-|\Delta x|/\sigma)$, where $\sigma$ is a free parameter. Assume large embedding dimension, $d \geq \text{rank}(\boldsymbol{M}^\star)$. Then in the continuum limit, the PCA-projected embeddings for the words $i \in \mathcal{S}$ are given by*

$$\lim_{|\mathcal{S}| \to \infty} \overline{\boldsymbol{w}}_i = \Big[a_1 \sin_{k_1}(k_1 x_i),\ a_2 \cos_{k_2}(k_2 x_i),$$
$$a_3 \sin_{k_3}(k_3 x_i),\ a_4 \cos_{k_4}(k_4 x_i),\ \dots\Big] \quad (7)$$

*where $\sin_{k_n}$ and $\cos_{k_n}$ are the mean-zero unit-normalized versions of $\sin$ and $\cos$ on $[-1, 1]$, the wavenumbers obey the self-consistent quantization condition*

$$k_n = \begin{cases} \dfrac{(n+1)\pi}{2} - \tan^{-1}(\sigma k_n), & n \text{ odd}, \\ \dfrac{n\pi}{2} + \tan^{-1}\left(\dfrac{k_n}{1 + \sigma(1+\sigma)k_n^2}\right), & n \text{ even}, \end{cases} \quad (8)$$

*such that $k_n < k_{n+1}$, and*

$$a_n = \sqrt{\frac{2\sigma}{1 + \sigma^2 k_n^2}}. \quad (9)$$

---

[3]The *periodized* kernel (i.e., sum over $n$) is the natural choice because, e.g., the total co-occurrence of "April" and "January" must account for not only the closest January to April, but also the following January, and the previous year's January, etc.

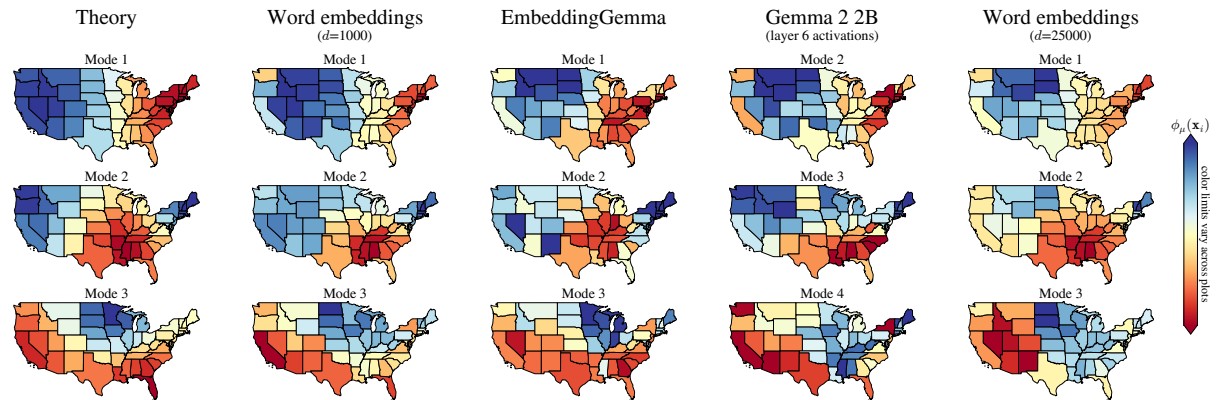

*Figure 3.* **Translation symmetry in geographic data yields top embedding modes with slow spatial variation.** We depict the top PCA modes (i.e., top eigenfunctions of the representational Gram matrix) for the 48 contiguous US states. In the leftmost column, we show the modes obtained from a theoretical model in which the co-occurrence probability of two states depends only on the Euclidean distance between their centroids, in geographic coordinates. Since the states do not lie on a perfect lattice, the theoretical eigenmodes are not exactly 2D plane waves; however, they still qualitatively exhibit slowly-varying oscillations. We compare to word embeddings, text embeddings, and LLM internal representations, and find good agreement. We observe that low-dimensional word embeddings more faithfully reflect the geographic semantic attributes (see Section 4 for an explanation as to why low-dimensional embeddings can be more robust to noise). See Appendix A for experimental details.

For historical years, Proposition 3 predicts that the feature manifold is an open loop whose components oscillate with historical time with non-integer frequencies. We validate this prediction in Figures 1, 2 and 11. See Appendix E.4 for the precise statement, its proof, and closed-form expressions for $\sin_{k_n}$ and $\cos_{k_n}$. The proof directly diagonalizes the co-occurrence statistics by exploiting special properties of the exponential kernel, enabling us to analytically handle the boundary effects.

### 3.1. Beyond the lattice approximation

The previous results describe the embedding geometry of words that are equally spaced on a semantic continuum. If the words are instead distributed unevenly (e.g., states on a map), we cannot analytically diagonalize $M^\star$. Nonetheless, we expect these results to provide a qualitatively correct description of the PCA modes: the top directions encode slow variations. We empirically validate this in Figure 3 using numerical diagonalization for theoretical predictions. The agreement serves as evidence that translation symmetry drives embedding geometry in two-dimensional semantic continuums (such as geographic locations) as well.

### 3.2. New predictions of the co-occurrence model

Propositions 1 and 3 predict that the projections of word representations onto any two principal components lie on Lissajous curves: parametric curves of the functional form $(x(t), y(t)) = (A\sin(at), B\sin(bt + \delta))$. In Figure 2 (left), we empirically confirm that Lissajous curves arise in the representation geometry of Wikipedia word embeddings. These results reveal the *monotonicity* of eigenmodes: since the amplitudes $a_n$ decrease with the frequencies $k_n$, the top

principal components encode the slowest Fourier modes. This explains why projecting the representations onto their top two directions yield circles and open loops, whereas 3D visualizations display "ripples" with higher extrinsic curvature (Gurnee et al., 2025).

A second prediction concerns how models may exploit the Fourier embedding geometry to do useful computations. We consider the coordinate decoding task in which one aims to predict $x_i$ from the representation vector $w_i$ using a linear probe. For example, Gurnee & Tegmark (2024) show that given LLM representations of events and places projected onto their top $r$ principal components, one may linearly decode the numerical value of historical years or the latitude and longitude of geographical locations. This property holds even when $r$ is relatively small. The following result explains this surprising phenomenon, assuming eigenmode monotonicity.

**Proposition 4** (Linear coordinate decoding (simplified))**.** *We work in the same setting as Proposition 1. Let $\overline{w}_{i,r} \in \mathbb{R}^r$ be the rank-$r$ PCA-projected embeddings, where $r \leq |\mathcal{S}|$. Given a linear probe $\Omega \in \mathbb{R}^{r \times D}$, define the decoding error*

$$\varepsilon^2(r) := \frac{\mathbb{E}_{i \in \mathcal{S}} \|\overline{w}_{i,r}^\top \Omega - x_i\|^2}{\mathbb{E}\|x_i\|^2}. \tag{10}$$

*If $\lambda_\mu$ is monotone decreasing with $|k_\mu|$, then for each $r$ there exists a linear probe $\Omega$ such that*

$$\lim_{|\mathcal{S}| \to \infty} \varepsilon^2(r) \leq \frac{6}{\pi^2} \left( \left(\frac{r}{\text{Vol}_D}\right)^{1/D} - \frac{\sqrt{D}}{2} \right)^{-1} \tag{11}$$

*where $\text{Vol}_D$ is the volume of the unit $D$-sphere. Thus the error scales asymptotically as $\varepsilon^2 \sim r^{-1/D}$.*

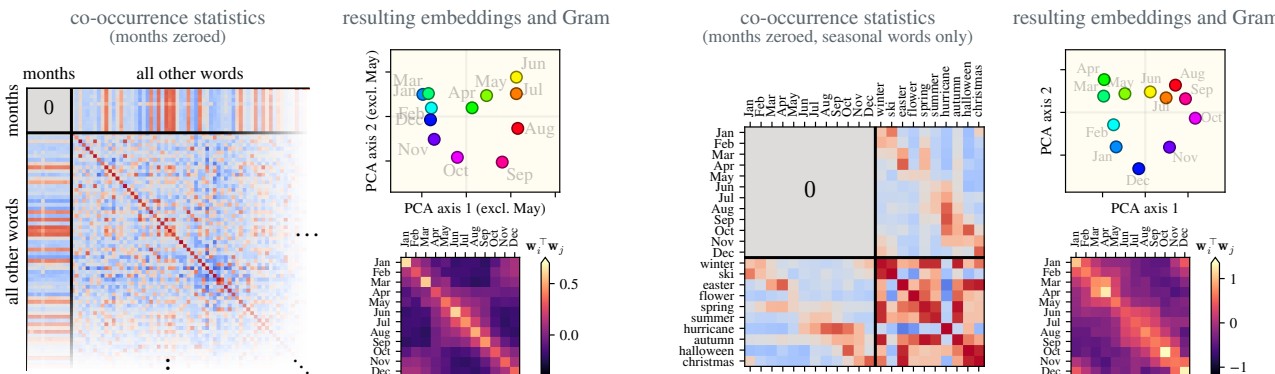

*Figure 4.* **Circular embedding of the months reconstructed without the month-month co-occurrences.** *Left*: we consider the English Wikipedia $M^*$ matrix ($V = 2.5 \times 10^4$), with the month-month co-occurrences ablated. Embeddings of dimension $d = 10^3 \ll V$ nonetheless exhibit a circular geometry (for a PCA over the months, excluding May) with the correct ordering, and lead to a Gram matrix that closely approximates the original month-month co-occurrences. That this is possible implies the existence of redundant time-of-the-year latent variables affecting the rest of the vocabulary. *Right:* we identify a subset of highly seasonal words and their phases (see Section C.1 for details), and show that just a handful of such seasonal words are sufficient to reconstruct the month ordering. See Appendix A for experimental details.

Put simply, Proposition 4 states that it is easy to linearly decode the spatial or temporal coordinates from the learned Fourier representations, even when they are low-passed (moderate $r$). We empirically validate the predicted scaling on embeddings trained on Wikipedia in Figure 2 (right). See Appendix E.5 for the full and precise statement, which handles finite lattices (no $|\mathcal{S}| \to \infty$ limit), and its proof. The proof idea is that since the representations encode Fourier modes of increasing frequency, the optimal rank-$r$ linear decoder computes the truncated Fourier series of $f(\boldsymbol{x}) = \boldsymbol{x}$, and the error is the missing high-frequency tail.

## 4. Collective effects control the embedding of space and time

Thus far, our analysis has focused on embeddings of maximal dimension $d = V$. In that case, the scalar product of two embedding vectors simply corresponds to the associated elements of the co-occurrence matrix. Our assumption above is that if the co-occurrence is restricted to the subset of words considered, e.g., the months of the year, it is a circulant matrix. Figure 4 (Left) shows however that the circular embedding of the months can be recovered if these two assumptions are broken simultaneously, by removing explicitly the co-occurrences of all the months in the PMI matrix, and by considering smaller embedding dimensions. In that new set-up, the Gram matrix of the month embeddings very nearly matches the raw co-occurrence statistics of the months, despite this reconstruction task utilizing only a small fraction of the eigenspectrum.

Such robustness of embedding properties to perturbation has already been observed in the context of analogical reasoning, e.g., King − Queen + Woman = Man (Mikolov et al., 2013). Remarkably, accurate analogy completions persist even when all sentences with a pair of words involved in the analogy (e.g., King, Queen) are explicitly removed from the corpus (Chiang et al., 2020). It was argued that this robustness is a consequence of a low-rank structure of the PMI matrix (Korchinski et al., 2025). We hypothesize that a similar effect is at play here and provide a mechanism for the emergence of an effectively low-rank structure.

Consider, as an example, the embedding of seasonality, i.e., time of the year (a similar argument can be made, e.g., for geography). We argue that many words are "seasonal," such that they occur more often when certain periods of the year are considered. For example, "ski" is associated to winter and will thus co-occur more frequently with, e.g., "December" than "August"; and the reverse can be said of "beach" or "hurricane." In this view, the time of the year $t$ is a latent variable that characterizes many words, which will globally affect the PMI co-occurrence matrix, leading to some large eigenvalues – a structure that is very robust to perturbations. This view is tested empirically in Figure 4 (Right), where we show that if the months of the year are complemented with highly seasonal words, the modified PMI in which all co-occurrences between months are removed still leads to circular months embeddings of some embedding dimension. This can be contrasted with highly *non-seasonal* words, like the numbers "one" through "seventeen," which do not enable seasonal reconstruction, as shown in Figure 15.

## 4.1. Co-occurrence arising from a shared continuous latent variable

To illustrate how recovery of the geometry is possible from co-occurrences with other seasonal words, we consider a simple setting where the collection of all seasonal words is governed by a periodic latent variable $t$ in the domain $[-T/2, T/2]$, which describes the time of year. Suppose each word $i$ has an intrinsic seasonal center $t_i$, uniformly distributed in $[-T/2, T/2]$. $t_i$ is a $D = 1$ version of the semantic lattice coordinates $\boldsymbol{x}_i$ introduced in Section 3, but we now consider the larger set—beyond, e.g., the calendar months—of seasonal words, which share statistical structure.

For word $i$, we assume its conditional probability of occurring at season $t$ is modulated by

$$P(i \mid t) = P(i)\big(1 + g(t - t_i)\big), \qquad (12)$$

where $g(t)$ is a symmetric, zero-mean function peaked at $t = 0$ that monotonically decreases towards $\pm T/2$. Normalization implies that $\sum_i P(i \mid t) = 1$. For a vocabulary of $N$ words taken to be equispaced in time, $t_i = \left(\frac{i}{N} - \frac{1}{2}\right)T$, normalization is obtained in the limit $N \to \infty$ e.g., if the $P(i)$'s are iid variables of finite variance and mean $1/N$.

Averaging over seasons, and assuming conditional independence, yields the joint marginal

$$P(i,j) = \int_{-T/2}^{T/2} \frac{dt}{T} P(i \mid t) P(j \mid t)$$
$$= P(i)P(j)\Big[1 + \tilde{K}(t_i - t_j)\Big], \qquad (13)$$

where $\tilde{K}$ is the circular autocorrelation

$$\tilde{K}(\Delta) = \frac{1}{T} \int_{-T/2}^{T/2} g(u)\, g(u + \Delta)\, du. \qquad (14)$$

Thus the PMI between two words depends only on their seasonal separation:

$$\text{PMI}(i,j) = \log \frac{P(i,j)}{P(i)P(j)} = \log\big(1 + \tilde{K}(t_i - t_j)\big) \quad (15)$$
$$=: K(t_i - t_j), \qquad (16)$$

which defines the kernel $K$ (approximately equal to $C(\cdot)$ in (3)). PMI = $\boldsymbol{K}$ is a circulant matrix.

### 4.1.1. DIAGONALIZATION OF THE PMI

Circulant matrices are diagonalized by discrete Fourier modes

$$\phi_{k(i)} = \frac{1}{\sqrt{N}} e^{2\pi i k t_i / T}, \qquad \boldsymbol{K}\phi_k = \mu_k \phi_k, \qquad (17)$$

where $\mu_k = \sum_{j=0}^{N-1} e^{2\pi i k j / N} K(t_j)$ is the discrete Fourier transform of $K$, and $k = 0, \pm 1, ..., \pm(N-1)/2$. For large

$N$ we have:

$$\lim_{N \to \infty} \mu_k / N = \int e^{2\pi i k t / T} K(t) dt. \qquad (18)$$

Because $K$ is smooth and even, its Fourier coefficients $\mu_k$ are real and decay rapidly with $|k|$. The eigenvector associated with $\mu_0$ is the constant mode, representing the global seasonal "mean." The next two eigenvectors, $\phi_1$ and $\phi_{-1} = \phi_1^*$, form a cosine–sine pair corresponding to a once-per-year oscillation. Subsequent eigenvectors encode higher-frequency seasonal variations.

**Robustness of embeddings to removal of the PMI elements:** Consider a fixed embedding dimension $d$, in the limit of large $N$. Because eigenvalues and the gap between them (or more precisely between degenerate pairs of eigenvalues) are proportional to $N$, any perturbation that affects a fixed number of entries of the PMI (such as the $12^2$ co-occurrence between months) will be much smaller than these gaps. Using the Weyl or Davis-Kahan theorem, we obtain that the top $d$ eigenvectors and thus the word embeddings are not perturbed in this limit.

## 4.2. Joint continuous and binary model

In the collective model just described, the only signal assumed to be present in the PMI matrix is "seasonality." In real data, there are many more signals present; e.g., signals that relate to other continuous latent variables such as "geography" or signals that lead to linear analogies of the type King − Man = Queen − Woman. In Appendix D, we build on Korchinski et al. (2025) to develop a model where both seasonality and correlations inducing linear analogies are present. We show that circular embedding geometry still emerges in this more realistic framework.

# 5. Discussion

Word embeddings, text embedding models, and LLMs all learn to represent time and space in a remarkable fashion: cyclic time exhibits circular geometry, historical time takes the form of open and curved one-dimensional manifolds, and geographic maps are represented as curved sheets. In addition, the high-dimensional "overtones" in these vectors enable linear probes to easily decode temporal and spatial coordinates. We have shown that this behavior emerges from a single unifying principle: pairs of words for times (or places) co-occur with a frequency that is dominated by their temporal (or spatial) separation. Importantly, this effect is collective, since there are many words that carry seasonal or geographical semantics. As a consequence, the learned representational geometry is remarkably robust to perturbations of the statistics.

We emphasize that the foundation of our theory lies in the low-order correlational statistics of the training data. We

prove our results in word embedding models, where closed-form training dynamics enable us to explicitly bridge the data statistics to the learned representations, but we find that these predictions carry over to a diverse set of larger sequence models. This suggests that the geometric structure of internal model representations is largely governed by task-agnostic and architecture-agnostic principles. Several recent works give evidence for this idea, including Shai et al. (2024); Prieto et al. (2026); Shai et al. (2026).

Though our experiments focus on model representations of time and space, a growing body of empirical evidence indicates that a zoo of such manifolds exists within language models, corresponding to concepts as varied as color, temperature, and age (Modell et al., 2025; Bhalla et al., 2026). Evidently, these models develop a form of perceptual grounding (e.g., internally representing hues as a color wheel) despite being trained exclusively on text. Assuming that these geometries are indeed driven by low-order text statistics, these findings suggest that the principles we develop in this paper may reveal the degree and character of such learned perceptual structure.

An intriguing question is whether such arguments play a role in neuroscience as well. In this respect, it is worth noting that grid cells—which encode two-dimensional space in the mammalian entorhinal cortex—display a firing pattern that can be interpreted as the interference of a small number of Fourier modes (Hafting et al., 2005; Dordek et al., 2016; Stachenfeld et al., 2017). Our work illustrates that seeking to predict the next position of the animal from a dataset of past trajectories (e.g., characterized as a succession of landmarks as in Raju et al. (2024)) will spontaneously create such representations, in either shallow or deep networks.

Ultimately, the representations are important because the model exploits their geometric structure during the forward-pass computation. For example, LLMs use the parallelogram geometry of linear analogies to complete relational tasks (Merullo et al., 2024), exploit Fourier structure for (modular) addition (Engels et al., 2025; Feucht et al., 2026), and use representational manifolds to reason about when to insert linebreaks (Gurnee et al., 2025). It may be useful to decompose model interpretability into two closely-related research programs: one which sharply predicts how representational geometry and computational primitives arise as a result of optimization pressures (as in this work), and one which describes how these pieces interconnect within the model to produce intelligent behavior.

## Limitations

Our theory is derived in the setting of word embedding algorithms. Remarkably, we demonstrated that it also captures many aspects of the geometry of representations in LLMs. The latter, however, have the ability to adapt to context;

for example, ambiguities may be removed. A very illustrative example is shown in Figure 14: if a prompt containing "May" provides disambiguating context, the representations of months become increasingly circular as they propagate through the forward pass. Building a theory to explain such phenomena would be very interesting.

In this work we consider the semantic relations between words that can be characterized by continuous attributes. This extends the previous work of Korchinski et al. (2025) that analyzed binary attributes and the associated emergence of parallelograms in representation space. Park et al. (2025b) have identified the emergence of geometric structure for hierarchical attributes that currently remains unexplained. Providing a global framework for all these properties would be desirable.

## Acknowledgements

We thank Blake Bordelon, Francesco Cagnetta, Alessandro Favero, Daniel Kunin, Eric Michaud, Jamie Simon, Antonio Sclocchi and the Simons Collaboration for useful conversations and mathematical insights. Y. Bahri thanks Andrew Lampinen, Yuxuan Li, and Eghbal Hosseini for conversations and Andrew Lampinen for feedback on a draft. D. J. Korchinski acknowledges financial support from the Natural Sciences and Engineering Research Council of Canada (NSERC PDF - 587940 - 2024). This work was supported by the Simons Foundation through the Simons Collaboration on the Physics of Learning and Neural Computation (Award ID: SFI-MPS-POL00012574-05), PIs Bahri and Wyart.

## Impact Statement

This paper presents work whose goal is to advance the field of Machine Learning. There are many potential societal consequences of our work, none which we feel must be specifically highlighted here.

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

# A. Experimental details

In this appendix we first provide more details about the code that produced the main text figures. We then summarize the experimental setup and evaluation details needed to reproduce and interpret our results. Our code is publicly available at https://github.com/dkarkada/symmetry-stats-repgeom.

## A.1. Main text figures

### A.1.1. FIGURE 1.

**(Top row.)** For our theoretical model, representation vectors are obtained by measuring the corpus statistics, constructing the matrix $M_{\mathcal{S}}^{\star}$, and using it to estimate the free parameters of the exponential kernel (see Figures 5 and 6). **(Middle row.)** For static word embeddings, the representation of "May" is perturbed due to its other meaning (i.e., to express possibility/permission/hope). To eliminate the effect of this polysemy, we exclude May from the computation of the PCA basis. **(Bottom row.)** For LLM internal representations, we use the prompt formulas given in Appendix A.3. The effect of polysemy on the representation of "May" is absent since the model is able to disambiguate from context (see Figure 14). Since Gemma 2 2B tokenizes digits separately, prompts for historical years that share the last digit or last two digits (e.g., 1735, 1835, and 1935) produce similar internal representations. This explains the bright off-diagonals in the Gram matrix.

**(Left column.)** For calendar months, we show *centered* Gram matrices (i.e., the Gram matrix of the mean-centered representation vectors). Since they have periodic BC, the constant mode is an eigenvector, so centering the embeddings simply subtracts a multiple of the all-ones matrix from the Gram matrix. **(Right column.)** For historical years, however, centering the embeddings changes the eigenbasis of the Gram matrix; to visually emphasize the Toeplitz-like nature of the uncentered eigenproblem, we show the uncentered Gram matrix instead. All geometry plots share the same color map: the red boundary is the year 1700 and the purple boundary is the year 2020. The geometric shape of the manifold is qualitatively insensitive to the chosen start and end years.

### A.1.2. FIGURE 2.

**(Left subplot, top.)** As before, the theoretical model is obtained by measuring the corpus statistics, constructing the matrix $M_{\mathcal{S}}^{\star}$, and using it to estimate the free parameters of the exponential kernel. The color map is as in Figure 1: the red boundary is the year 1700 and the purple boundary is the year 2020. **(Left subplot, bottom.)** The local "kinks" in the empirical Lissajous curves in the cyan and blue region correspond to the years of World War I and World War II. The preponderance of Wikipedia articles discussing the events of those wars (and other wars) weakly breaks the time translation symmetry, distorting the representational manifold.

**(Right subplot.)** We restrict our experiment to the word embeddings for the years 1900-2020. (We make this choice simply for variety; choosing a different set of years does not meaningfully affect the results.) For each rank-$r$ projection of the embeddings, we run 100 trials; each trial we randomly select 60 embeddings for a training set, reserving the remaining 60 for a test set, and train a linear probe using ridgeless linear regression to decode the numerical year. We depict both the trial-wise mean of the train/test errors as markers, and the standard deviation as error bars. The colored dotted lines are simply visual aids connecting the markers. The faint vertical dotted line denotes the so-called "interpolation threshold" at which the learning problem transitions from underparameterized to overparameterized.

In practice, one may estimate the optimal linear probe using nonzero ridge regularization. Although this improves test performance, the purpose of Figure 2 is to depict how the optimal *training* error decays in the underparameterized regime. Therefore we relegate the figure depicting the regularized performance to Figure 13.

### A.1.3. FIGURE 3.

Since the embeddings lie on a two-dimensional manifold (since $D = 2$ for geography), they are difficult to visualize; instead, we plot the eigenmodes directly. Across the various subfigures, the ranges of the color maps do not coincide; we set the upper and lower boundaries of each color map separately to maximize visual clarity since the different eigenmodes have different extremal values.

**(Column 1.)** The theoretical model is obtained as follows. Let $(x_i, y_i)$ be the latitude and longitude of the centroid of state $i$. Let $d_{ij} := \sqrt{(x_i - x_j)^2 + 0.78(y_i - y_j)^2}$. The factor 0.78 is approximately the aspect ratio of one square degree of latitude/longitude in the continental US. Therefore, $d_{ij}$ approximately measures the loxodromic distance between state $i$ and $j$. We use the theoretical co-occurrence model $M_{\mathcal{S}(ij)}^{\star} = 10 \exp(-d_{ij}/20)$, where the parameters 10 and 20 were chosen by visual inspection; the predicted eigenmodes are qualitatively insensitive to these choices. We then numerically

diagonalize $M_{\mathcal{S}}^{\star}$ to obtain the embeddings.

**(Columns 3, 4.)** We use the prompt formula given in Appendix A.3. For the Gemma 2 2B plots, we omit the first eigenmode because it reflects tokenization rather than semantics: it activates strongly for states with two tokens in their name (e.g., West Virginia, New York, North Dakota).

### A.1.4. FIGURE 4.

In the left panel, we depict an experiment in which we set $P_{ij} = P_i P_j$ for every $i, j \in \mathcal{S}$, the set of target words. The effect is to set $M_{\mathcal{S}}^{\star} = 0$. The embedding geometry (extracted in the usual manner via SVD, as described in Section 2.1) again excludes May from the PCA computation. The Gram matrix is computed from the centered embeddings. In the right panel, we obtain word embeddings by factorizing the submatrix of $M^{\star}$ containing only the 12 months and the 10 chosen "seasonal words," with the month-month co-occurrences still satisfying $M_{\mathcal{S}}^{\star} = 0$. We do *not* exclude May from the PCA, and the Gram matrix remains centered.

### A.2. Word embeddings

**Corpus.** We train all word embedding models on the November 2023 dump of English Wikipedia, downloaded from `https://huggingface.co/datasets/wikimedia/wikipedia`. We lowercase the corpus, remove all numerals with commas or decimal points, replace all non-alphanumeric characters with whitespace, and tokenize by splitting on whitespace. We discard all articles with fewer than 200 tokens, leaving 3.37 million articles.

We use a vocabulary consisting of the $V = 25000$ most frequently appearing words. This automatically includes the words corresponding to the continuous concepts we probe (months, years, and US states). We discard out-of-vocabulary words from the corpus; our robustness checks indicated that it does not practically matter whether out-of-vocabulary words are removed or simply masked. Ultimately, the training corpus comprises 2.72 billion tokens.

**Co-occurrence probabilities.** Given hyperparameters $L$ (representing the context window size) and $f(\cdot)$ (representing a distance-based reweighting function), the co-occurrence (skip-gram) distribution $P_{ij}$ of corpus $\mathcal{C}$ is defined

$$P_{ij} = \frac{1}{Z} \sum_{\nu} \delta_{\mathcal{C}[\nu],i} \sum_{d=1}^{L} f(d) \left( \delta_{\mathcal{C}[\nu+d],j} + \delta_{\mathcal{C}[\nu-d],j} \right) \tag{19}$$

where $\mathcal{C}[\nu]$ is the $\nu$-th token in the corpus, $\delta$ is the Kronecker delta, and $Z$ is simply the normalizing constant. We choose $L = 16$ and $f(d) = L + 1 - d$ simply for consistency with well-known word embedding algorithms; we find that our results are not strongly sensitive to these choices. In particular, these correspond to a hyperparameter scheme similar to the original `word2vec` algorithm, where co-occurring word pairs are collected from symmetrical contexts whose window size varies uniformly randomly between 1 and $L$. This dynamic context window has the aggregate effect of downweighting the co-occurrence probability of words exactly as our $f(d)$. This decreases the co-occurrence probability of word pairs that are typically separated by many words, compared to e.g., common bigrams. The unigram probabilities are defined $P_i \coloneqq \sum_j P_{ij}$.

**Training.** We first construct the target matrix $M^{\star}$ from the co-occurrence statistics according to Equation (1). We then obtain $W$ by diagonalizing $M^{\star}$ and evaluating Equation (2). See Appendix B for a complete discussion.

### A.3. Gemma 2 2B internal representations

We utilize the two-billion-parameter Gemma 2 2B model (`google/gemma-2-2b`) (Gemma Team, 2024). It has a vocabulary of 256K tokens and the model hidden size is $d = 2304$. Using the `TransformerLens` library (Nanda & Bloom, 2022) and loading weights from the Hugging Face Hub, we extract residual-stream activations from the end of each transformer block (`blocks.{l}.hook_resid_post`) for layers $l \in \{0, \dots, 25\}$. We analyze the activations for three semantic categories: calendar months (e.g., "January"), historical years (e.g., "1776"), and US states (e.g., "New York"). For each string $x$, we construct a prompt using the templates below:

| | |
|---|---|
| Calendar months: | "The month of the year is $x$" |
| Historical years: | "In the year $x$" |
| US states: | "The location of the US state $x$" |

We retain the activation vector corresponding to the final token position $T - 1$, where $T$ is the tokenized prompt length.

### A.4. EmbeddingGemma representation vectors

We utilize the 308 million parameter text embedding model `EmbeddingGemma` (Vera et al., 2025) via the `huggingface` library. `EmbeddingGemma` is derived from the `Gemma3` model family and yields fixed-length vector representations of text. We obtain the vectors using the SentenceTransformers module (Reimers & Gurevych, 2019) and query for vectors with the default size (768-dimensional) and unit normalization for the semantic category US states. The queried sentences are identical to those in Appendix A.3, namely:

$$\text{US states:} \quad \text{``The location of the US state } x\text{''}$$

with string $x$ corresponding to each of the 48 contiguous states. The top PCA modes of these representations are shown in Figure 3.

# B. Review of word embedding algorithms

In this appendix we review recent results showing that word embedding algorithms are well-approximated by spectral methods. We first review the mechanism by which symmetric word embeddings trained via self-supervised gradient-based algorithms learn the spectral decomposition of the positive part of the co-occurrence matrix $M^\star$. We then discuss how the PCA geometry of *asymmetric* word embeddings recovers instead the matrix absolute value $|M^\star|$. Finally, we discuss the relation between $M^\star$ and the well-known pointwise mutual information (PMI) matrix.

## B.1. Symmetric word embeddings from $M^\star$

In the main text, we use $W$ to denote the trained embeddings. Since we are now discussing optimization, we change notation, using $W$ to instead refer to the variable being optimized, and $\hat{W}$ to denote the optimal value.

Let the co-occurrence (skip-gram) distribution $P_{ij}$ and the unigram distribution be defined as in Section A.2. Let $V$ be the number of words in the vocabulary, and let $W \in \mathbb{R}^{V \times d}$ be a trainable weight matrix whose $i^{\text{th}}$ row is the $d$-dimensional embedding vector for word $i$. Word embedding algorithms such as word2vec aim to imbue $W$ with semantic structure so that the inner products between word embeddings capture semantic similarity. In particular, these self-supervised algorithms iterate over the text corpus, aligning the embeddings of frequently co-occurring words and repelling unrelated words.

Karkada et al. (2025) show that the following symmetric matrix factorization problem well-approximates word2vec, producing word embeddings of comparable linear algebraic structure and semantic quality:

$$\hat{W} = \arg\min_{W} \|WW^\top - M^\star\|_{\text{F}}^2, \tag{20}$$

where the target matrix $M^\star$ is defined as in Equation (1):

$$M^\star_{(ij)} := \frac{P_{ij} - P_i P_j}{\frac{1}{2}(P_{ij} + P_i P_j)} \tag{21}$$

$$= 2\frac{\frac{P_{ij}}{P_i P_j} - 1}{\frac{P_{ij}}{P_i P_j} + 1} \tag{22}$$

$$\approx \log\left(\frac{P_{ij}}{P_i P_j}\right). \tag{23}$$

Intuitively, $M^\star$ encodes the normalized excess co-occurrences of word pairs; $M^\star_{(ij)} > 0$ when words $i$ and $j$ co-occur more frequently than if the words were drawn independently, and vice versa. Some additional properties of $M^\star$ include that its elements are bounded ($-2 \le M^\star_{(ij)} \le 2$) and that it is real symmetric (and therefore has a real-valued spectral decomposition $M^\star = \Phi\Lambda\Phi^\top$). The approximation in the last line shows that $M^\star$ is approximately equal to the *pointwise mutual information* (PMI) matrix, elementwise; we derive and discuss it in Appendix B.4.

Karkada et al. (2025) show that running the word2vec algorithm is approximately equivalent to running gradient descent on the objective in Equation (20), which is known to converge to $\hat{W}$ with high probability. By the Eckart-Young-Mirsky theorem, the trained embeddings must have the structure

$$\hat{W} = \Phi_{(\cdot, :d)}\sqrt{\Lambda_{(:d, :d)}} \tag{24}$$

up to semantically-irrelevant right orthogonal transformations. In other words, $W$ learns to model the $d$ largest positive eigenmodes of $M^\star$.

A limitation of this analysis is that it is restricted to symmetric factorizations. This prevents $W$ from ever learning eigenmodes of $M^\star$ with negative eigenvalues. To be precise, let us define the positive semidefinite and negative semidefinite components of $M^\star$, $M^+$ and $M^-$ respectively, to be the unique matrices satisfying

$$M^\star = M^+ - M^- \qquad M^+ \succeq 0 \qquad M^- \succeq 0 \qquad M^+ M^- = 0. \tag{25}$$

Then, even if $WW^\top$ has no rank constraint, one may only achieve $WW^\top = M^+ \ne M^\star$ in general.

**B.2. Asymmetric word embeddings from $M^\star$**

To circumvent this limitation, we consider the asymmetric setting

$$(\hat{\boldsymbol{W}}, \hat{\boldsymbol{W}}') = \arg\min_{\boldsymbol{W}, \boldsymbol{W}'} \|\boldsymbol{W}\boldsymbol{W}'^\top - \boldsymbol{M}^\star\|_{\mathrm{F}}^2, \tag{26}$$

where $\boldsymbol{W}$ is the word embedding matrix and $\boldsymbol{W}'$ is known as the context embedding matrix. Unlike the case of symmetric factorization, where global minimizers are all related by right orthogonal transformations, the asymmetric objective has a substantially larger invariance: $(\hat{\boldsymbol{W}}, \hat{\boldsymbol{W}}') \equiv (\hat{\boldsymbol{W}}\boldsymbol{A}, \hat{\boldsymbol{W}}'\boldsymbol{A}^{-\top})$ for any invertible matrix $\boldsymbol{A}$. Therefore the geometric structure of the learned word embeddings $\hat{\boldsymbol{W}}$ is identifiable only up to arbitrary linear transformations.

This is a highly underdetermined problem. To simplify the analysis of the training dynamics, we consider a special initialization scheme: $\boldsymbol{W}(0)$ initialized with i.i.d. Gaussian elements as usual, and $\boldsymbol{W}'(0) = \boldsymbol{Q}\boldsymbol{W}(0)$ with random orthogonal matrix $\boldsymbol{Q}$. Then, exploiting the well-known conservation law of gradient flow on deep linear networks, i.e., $\frac{d}{dt}(\boldsymbol{W}^\top\boldsymbol{W} - \boldsymbol{W}'^\top\boldsymbol{W}') = 0$, we conclude that the right singular vectors of $\boldsymbol{W}$ and $\boldsymbol{W}'$ must agree throughout training (and likewise for the singular values). Consequently, let us denote the SVD of the optimal $\boldsymbol{W}$ as $\hat{\boldsymbol{W}} = \hat{\boldsymbol{\Phi}}\hat{\boldsymbol{S}}$ and the SVD of the optimal $\boldsymbol{W}'$ as $\hat{\boldsymbol{W}}' = \hat{\boldsymbol{\Phi}}'\hat{\boldsymbol{S}}$, where without loss of generality we assume the right singular vectors are simply identity. Invoking Eckart-Mirsky-Young again, and re-indexing the modes of $\boldsymbol{M}^\star$ to be non-increasing in the *magnitude* of its eigenvalues, we find

$$\hat{\boldsymbol{W}}\hat{\boldsymbol{W}}'^\top = \hat{\boldsymbol{\Phi}}\hat{\boldsymbol{S}}^2\hat{\boldsymbol{\Phi}}'^\top \tag{27}$$

$$= \boldsymbol{\Phi}_{(\cdot,:d)}\boldsymbol{\Lambda}_{(:d,:d)}\boldsymbol{\Phi}_{(\cdot,:d)}^\top \tag{28}$$

$$= \boldsymbol{\Phi}_{(\cdot,:d)}\sqrt{|\boldsymbol{\Lambda}_{(:d,:d)}|}\sqrt{|\boldsymbol{\Lambda}_{(:d,:d)}|}\boldsymbol{D}\boldsymbol{\Phi}_{(\cdot,:d)}^\top \tag{29}$$

where $\boldsymbol{D} := \mathrm{sign}(\boldsymbol{\Lambda})$ is the diagonal matrix containing the signs of the modes of $\boldsymbol{M}^\star$. Clearly, the learned solutions are

$$\hat{\boldsymbol{W}} = \boldsymbol{\Phi}_{(\cdot,:d)}\sqrt{|\boldsymbol{\Lambda}_{(:d,:d)}|} \tag{30}$$

$$\hat{\boldsymbol{W}}' = \left(\boldsymbol{\Phi}_{(\cdot,:d)}\boldsymbol{D}\right)\sqrt{|\boldsymbol{\Lambda}_{(:d,:d)}|}. \tag{31}$$

Since we will no longer discuss optimization, we revert to our previous notation where $\boldsymbol{W}$ and $\boldsymbol{W}'$ refer to the optima. This yields the expression given in Equation (2). However, the PCA geometry of the word embeddings is given by the gram matrix $\boldsymbol{W}\boldsymbol{W}^\top$, not the model matrix $\boldsymbol{W}\boldsymbol{W}'^\top$:

$$\boldsymbol{W}\boldsymbol{W}^\top = \boldsymbol{\Phi}_{(\cdot,:d)}|\boldsymbol{\Lambda}_{(:d,:d)}|\boldsymbol{\Phi}_{(\cdot,:d)}^\top \tag{32}$$

Indeed, if the embedding dimension is sufficiently large, $d \geq \mathrm{rank}\,\boldsymbol{M}^\star$, the gram matrix factors the matrix absolute value of $\boldsymbol{M}^\star$, defined $|\boldsymbol{M}^\star| := \boldsymbol{M}^+ + \boldsymbol{M}^-$:

$$\boldsymbol{W}\boldsymbol{W}^\top = |\boldsymbol{M}^\star|. \tag{33}$$

**B.3. Weaker version of Assumption 3.1**

Note that if $\boldsymbol{M}^\star$ is positive semi-definite, then $\boldsymbol{M}^- = \boldsymbol{0}$, and the result Equation (32) is exactly the same as in the symmetric case. In particular, the gram matrix exactly recovers $\boldsymbol{M}^\star$. This convenience is why we include the PSD assumption in Assumption 3.1.

However, this assumption is violated in real data. Indeed, the empirical spectrum of $\boldsymbol{M}^\star$ on Wikipedia has a unimodal distribution peaked near $\lambda = 0$, and $\mathrm{rank}\,\boldsymbol{M}^+ \approx \mathrm{rank}\,\boldsymbol{M}^-$ (Figure 7). Therefore, in Figures 1 and 2, the exponential kernel estimated by fitting to $\boldsymbol{M}^\star$ does not, strictly speaking, represent the kernel that is actually factored (which should be measured from $|\boldsymbol{M}^\star|_{\mathcal{S}}$ instead). This explains why we typically do not accurately predict the absolute scale of the mode amplitudes. Despite this, the predictions of Corollary 2 and Proposition 3 appear to correctly predict both the relative amplitudes between modes and the wavenumbers.

To develop theoretical results that more faithfully model real data, we propose the following drop-in replacement for Assumption 3.1:

**Assumption B.1. Translation symmetry of $|M^\star|$ with kernel $C(\cdot)$.** Define $M^\star$ as in Equation (1), and define its unique decomposition into PSD and NSD components $M^\star = M^+ - M^-$ as in Equation (25). Let $C^+(\cdot)$ and $C^-(\cdot)$ both be well-behaved functions. For any two words $i, j \in \mathcal{S}$, assume

$$M^+_{(ij)} = C^+(\text{dist}(\boldsymbol{x}_i, \boldsymbol{x}_j)) \qquad \text{and} \qquad M^-_{(ij)} = C^-(\text{dist}(\boldsymbol{x}_i, \boldsymbol{x}_j)) \tag{34}$$

so that

$$|M^\star|_{(ij)} = C^+(\text{dist}(\boldsymbol{x}_i, \boldsymbol{x}_j)) + C^-(\text{dist}(\boldsymbol{x}_i, \boldsymbol{x}_j)) \tag{35}$$
$$= C(\text{dist}(\boldsymbol{x}_i, \boldsymbol{x}_j)) \tag{36}$$

where $C(\cdot) := C^+(\cdot) + C^-(\cdot)$ is the *co-occurrence convolution kernel*.

Assumption B.1 implies that both $|M^\star|_\mathcal{S}$ and $M^\star_\mathcal{S}$ have a translation symmetry (with different convolution kernels). Note that Assumption B.1 strictly softens Assumption 3.1, which corresponds to the special case $M^- = 0$. In addition, Assumption B.1 is well-supported in both the empirical data (Figures 5 and 6) and in the collective model of Section 4.1. In particular, if the function $g$ is large enough in magnitude, the non-linear effects associated with the logarithm in Equation (15) can lead to both positive and negative values $\mu_k$.

However, estimating the kernel $C^+ + C^-$ from the raw data statistics requires computing $|M^\star|_\mathcal{S}$. This involves a global spectral transformation of the co-occurrence probabilities for *all* words. This contrasts estimating the kernel $C^+ - C^-$ from $M^\star_\mathcal{S}$, which only requires knowledge of co-occurrence probabilities and unigram probabilities for the words in $\mathcal{S}$. In this sense, the approximate Assumption 3.1 is easier to use than Assumption B.1. For this reason, we choose to use Assumption 3.1 when producing theoretical predictions in Figures 1 and 2.

### B.4. Relation to PMI matrix

Early word and document embedding algorithms obtained low-dimensional embeddings by explicitly constructing some target matrix and employing a dimensionality reduction algorithm. For instance, latent semantic analysis Deerwester et al. (1990) constructed document embeddings the SVD of from correlations between terms and documents, while Turney (2004) used the SVD of word pair relations – the latter produced embeddings solving linear analogies. For words, one popular choice of target matrix was the *pointwise mutual information* (PMI) matrix (Church & Hanks, 1990), defined

$$\text{PMI}_{(ij)} = \log \frac{P_{ij}}{P_i P_j}. \tag{37}$$

Later, Levy & Goldberg (2014) showed that PMI is the rank-unconstrained minimizer of the `word2vec` objective. To see the relation between PMI and the normalized co-occurrence $M^\star$, let us first write

$$\log\left(\frac{P_{ij}}{P_i P_j}\right) = \log(1 + \Delta(x_{ij})), \tag{38}$$

where the function $\Delta(x)$ represents the fractional deviation away from independent word statistics ($P_{ij} = P_i P_j$), in terms of some small parameter $x$ of our choosing (so that $\Delta(0) = 0$). This setup allows us to Taylor expand quantities of interest around $x = 0$. A judicious choice will produce terms that cancel the $-\frac{1}{2}\Delta^2$ that arises from the Taylor expansion of $\log(1 + \Delta)$, leaving only third-order corrections. One such example is $\Delta(x) = 2x/(2 - x)$, which yields

$$\text{PMI} = \log\left(1 + \frac{2x}{2 - x}\right) = x + \frac{x^3}{12} + \frac{x^5}{80} + \cdots \tag{39}$$

and has the solution

$$x_{ij} = \frac{P_{ij} - P_i P_j}{\frac{1}{2}(P_{ij} + P_i P_j)} = M^\star_{(ij)}. \tag{40}$$

This calculation reveals that $M^\star$ is a very close approximation to the PMI matrix; the leading correction is third order. For this reason, we use the PMI matrix in the theoretical analysis of Section 4. However, $M^\star$ is empirically much friendlier to least squares approximation because its elements are bounded ($-2 \leq M^\star_{(ij)} \leq 2$). Indeed, Levy et al. (2015) observed that

word embeddings obtained by least squares factorization of the PMI matrix perform poorly on downstream tasks. One can mitigate this problem by instead factorizing a regularized variant of the PMI matrix:

$$\mathrm{PMI}_\epsilon = \log\left(\frac{P_{ij}}{P_i P_j} + \epsilon\right) \tag{41}$$

for some hyperparameter $\epsilon$.

# C. Additional empirical evidence

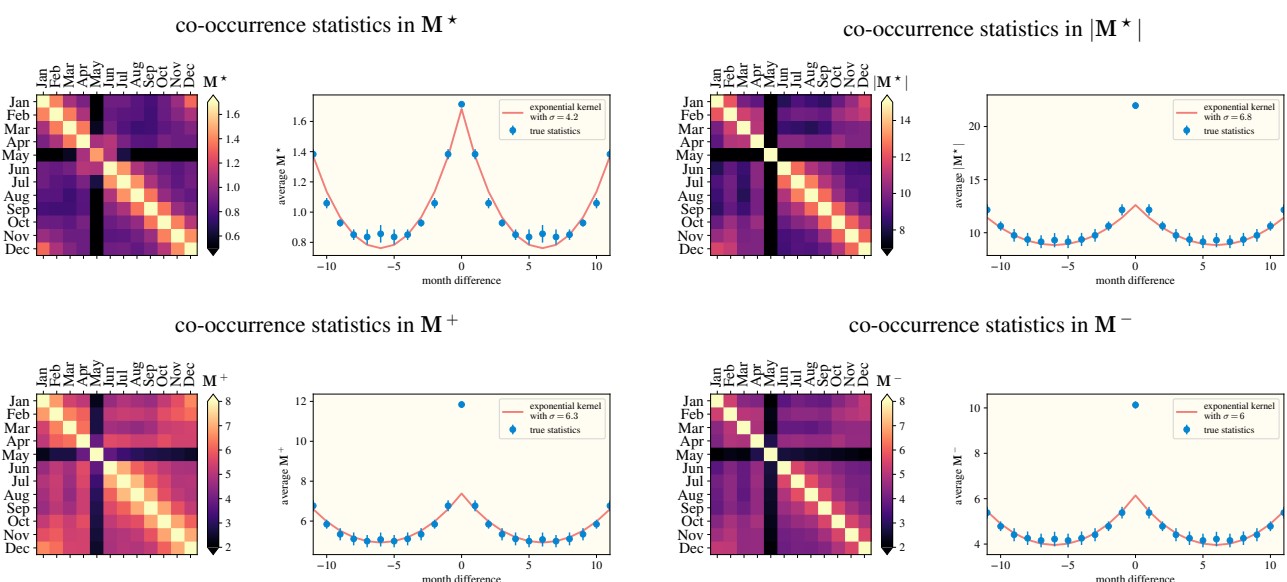

*Figure 5.* **Translation symmetry in month co-occurrence statistics.** The four main quadrants of the plot correspond to the matrices $\boldsymbol{M}^\star = \boldsymbol{M}^+ - \boldsymbol{M}^-$, $|\boldsymbol{M}^\star| = \boldsymbol{M}^+ + \boldsymbol{M}^-$, $\boldsymbol{M}^+$, and $\boldsymbol{M}^-$, as defined in Equation (25). For each, on the left we display the submatrix of calendar months and observe the circulant structure indicative of time translation symmetry. On the right, we plot the normalized co-occurrence as a function of the time interval between months, with bars indicating one standard deviation of variation across months. We find excellent agreement with the periodized exponential kernel, except along the main diagonal. We argue that this discrepancy does not affect eigenvector ordering, since adding $\kappa \boldsymbol{I}$ to any matrix $\boldsymbol{A}$ only shifts each eigenvalue by a constant $\kappa$ (leaving the eigengaps unchanged). We use the periodized kernel $C(\Delta x) = \sum_{n \in \mathbb{Z}} \exp(-|\Delta x + 2n|/\sigma)$ because, e.g., a co-occurrence of April and January may refer not only to the closest January to April, but also with the following January, and with the previous year's January, etc.

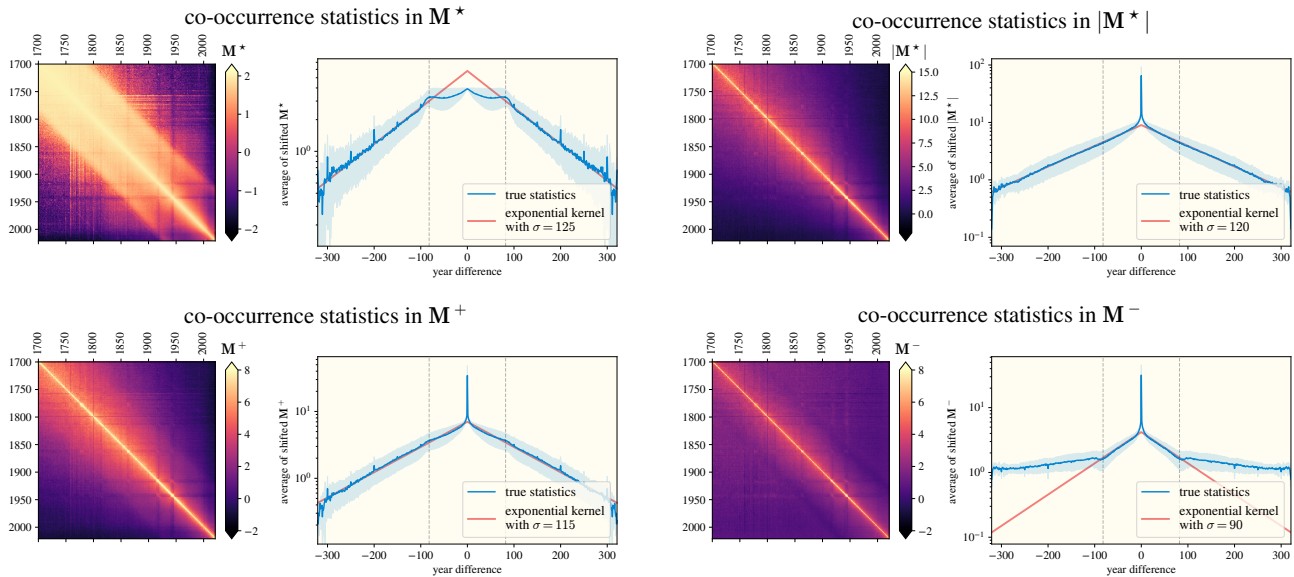

*Figure 6.* **Translation symmetry in year co-occurrence statistics.** Exactly analogous to Figure 5. Since $\boldsymbol{M}^\star$ takes negative values, we first shift the empirical kernel by a constant before fitting the exponential kernel $C(\Delta x) = \exp(-|\Delta x|/\sigma)$. This is equivalent to fitting a shifted exponential kernel to the statistics. We expect that the analysis in Proposition 3 remains valid, since PCA projection will annihilate the constant mode.

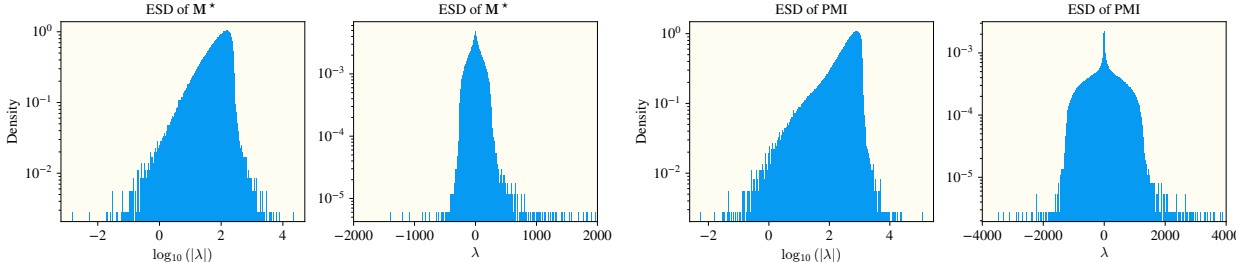

*Figure 7.* **Empirical spectral distribution of $M^\star$ and PMI.** We show the histogram of eigenvalues of both $M^\star$ (left two plots) and the PMI matrix (right two plots). For each, we plot both the logarithmic density $p(t)$ for $t := \log(|\lambda|)$ as well as $p(\lambda)$ directly. The latter shows that the spectrum is roughly symmetric, with the PSD and NSD components being of comparable rank (rank $M^+ \approx$ rank $M^-$ as defined in Equation (25)). The former displays a linear growth, indicating a spectral bulk with no divergence at zero, followed by a steep drop-off, which may be interpreted as a bulk spectral edge. Many eigenvalues exist beyond the spectral edge, suggesting that these are semantic signals.

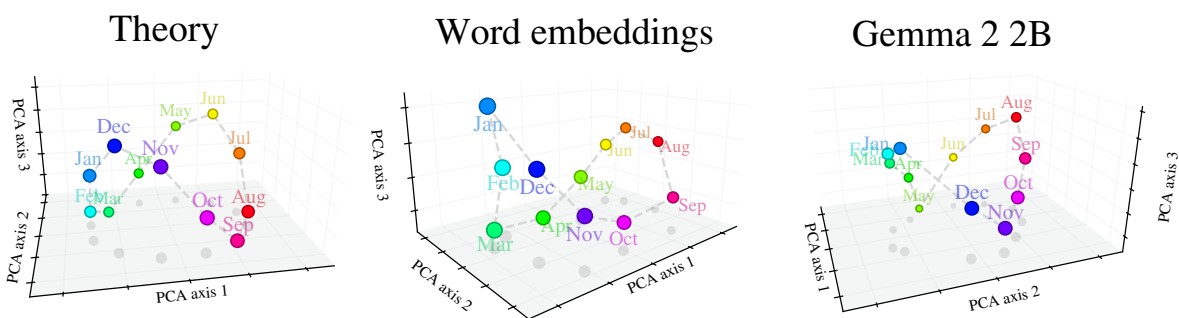

*Figure 8.* **Calendar Pringles.** Visualizing the calendar month representations in 3D, we observe the saddle-like geometry reported in prior work. This is a direct consequence of the Fourier geometry: the third mode encodes the first overtone above the fundamental frequency.

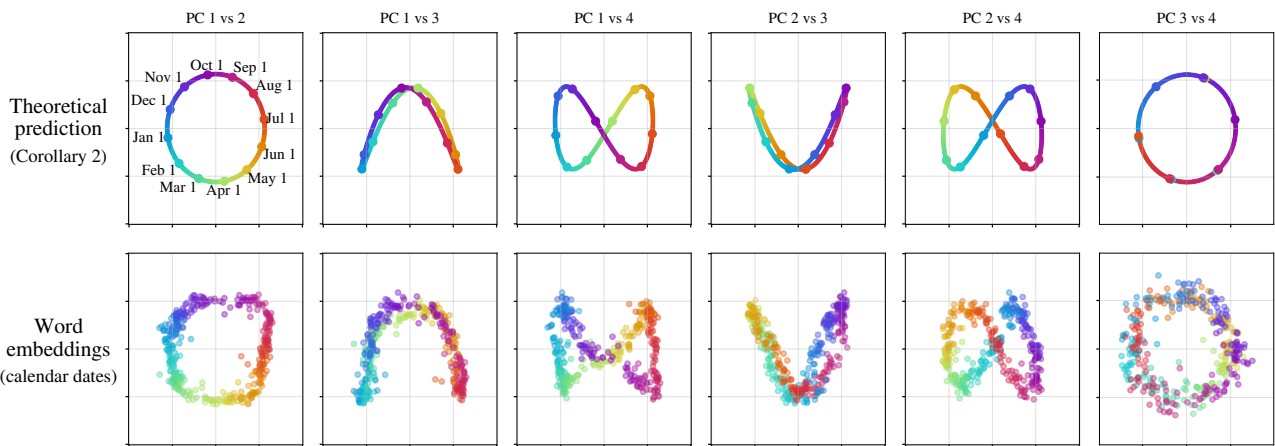

*Figure 9.* **Lissajous curves for calendar dates.** We tokenize the corpus to treat date phrases such as "January 1" as a single word. The resulting representational manifold for the 365 calendar dates displays Lissajous curves whose amplitudes, phases, and frequencies are analytically predicted by Corollary 2.

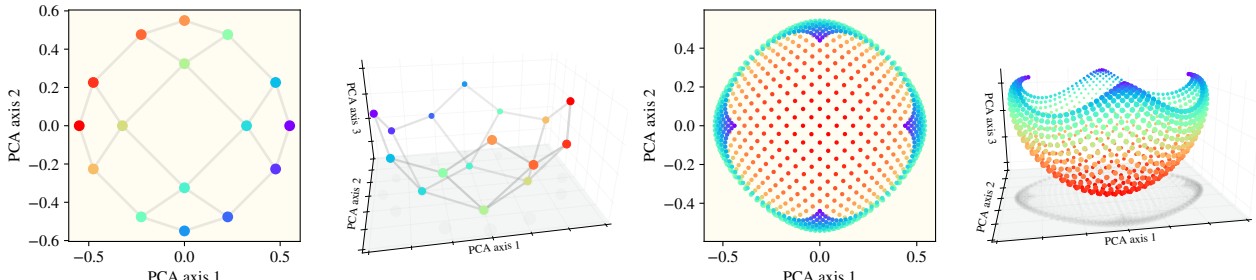

*Figure 10.* **Embedding geometry for 2-dimensional lattices.** We show the embedding geometry for 2D lattices with open BC and exponential kernel. The only difference between the left two and right two plots is that the lattice spacing is smaller on the right ($4 \times 4$ grid vs. $31 \times 31$ grid). For the denser grid (closer to the continuum limit), the cross sections of the embedding manifold have the same horseshoe-like geometry seen in the 1-dimensional case. In both cases, the center of the grid is faithfully reproduced in representation space, while the edges are distorted. This matches the results reported in Park et al. (2025a); we hypothesize that LLMs solve their in-context learning task (random walks on a grid) by computing co-occurrence statistics in-context using a matrix-factorization-like mechanism. This hypothesis is supported by a large literature explaining how transformers learn Markov models by computing the latent variables in-context (Bietti et al., 2023; Edelman et al., 2024; Nichani et al., 2024; Chen et al., 2024).

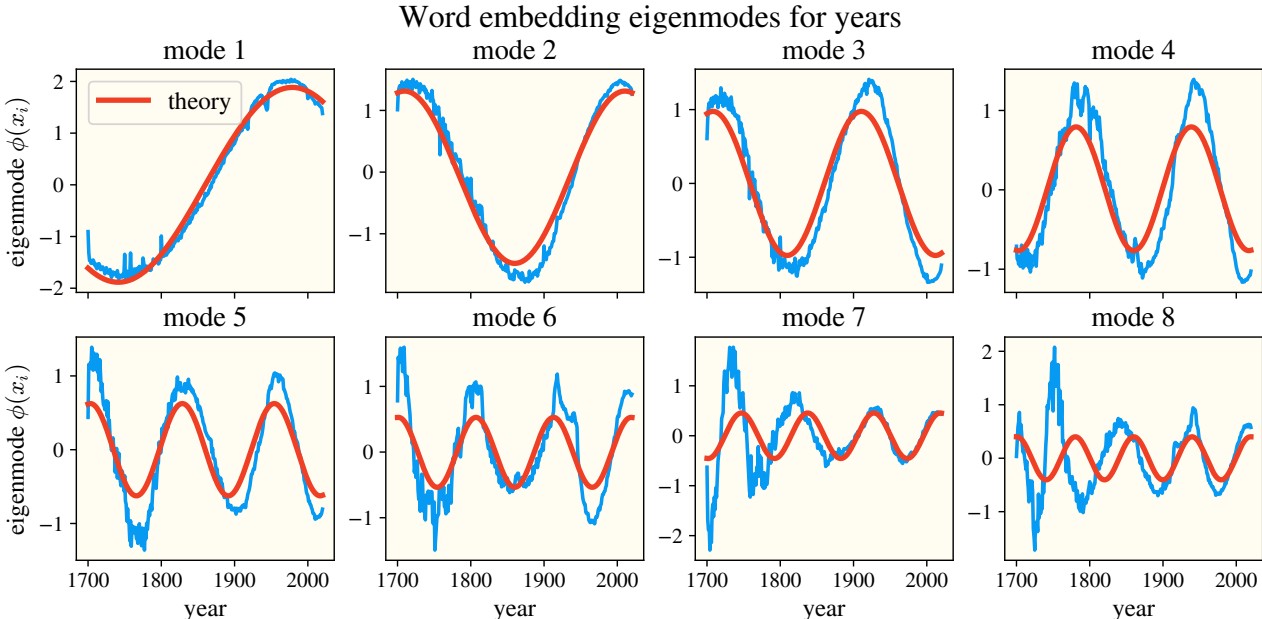

*Figure 11.* **Eigenmodes of history.** We provide an alternate visualization of the result in the left panel of Figure 2. We plot the eigenmodes of the embeddings for historical years and compare with our theoretical predictions. The Lissajous curves in Figure 2 are obtained by choosing two of these modes, $\mu$ and $\nu$, and plotting the parametric curve $(x(t), y(t)) = (\phi_\mu(t), \phi_\nu(t))$.

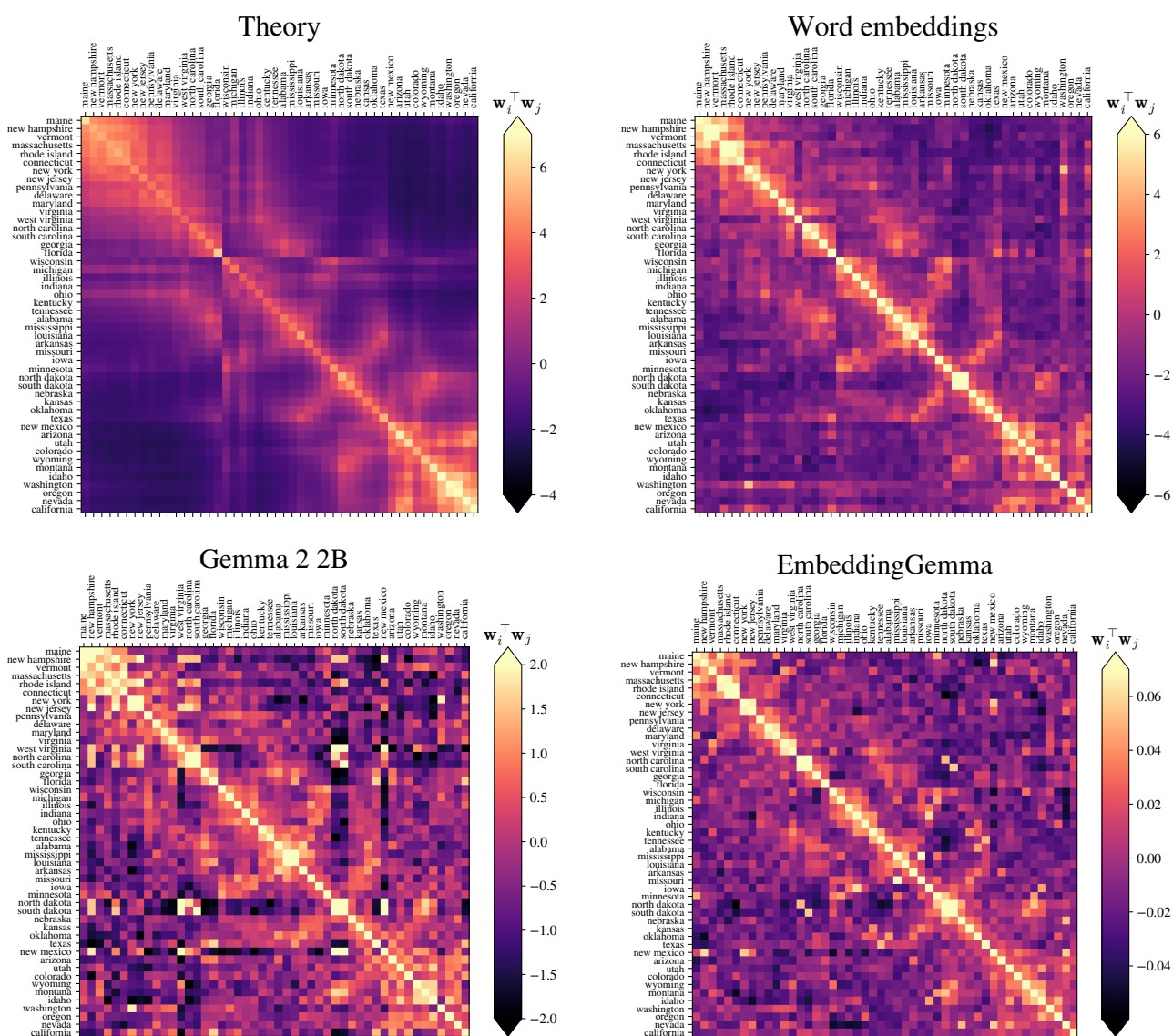

*Figure 12.* **Gram matrices for US geography.** We display the centered Gram matrices for model representations of the 48 contiguous US states, across the various models shown in Figure 3. The theoretical Gram matrix depends only on geographic distance, according to Appendix A.1.3. The empirical Gram matrices evidently contain other semantic signals; nonetheless, the geographic structure dominates, as evidenced in Figure 3.

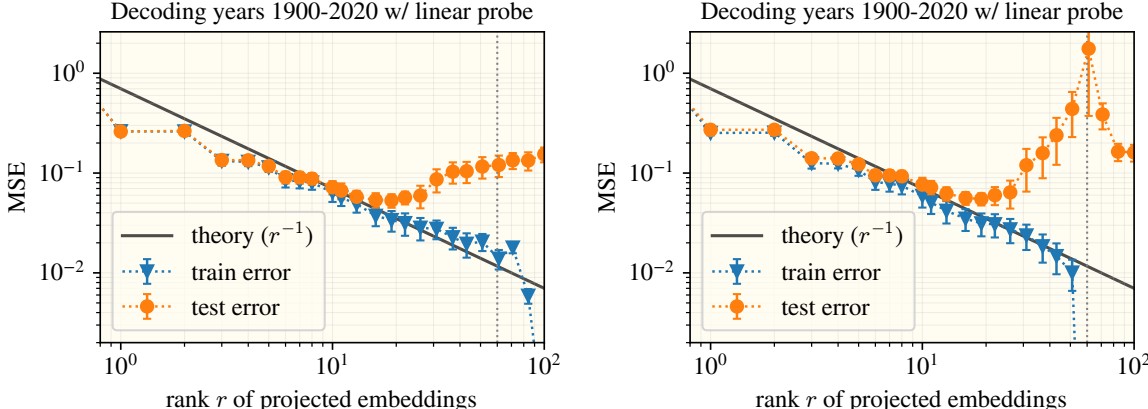

*Figure 13.* **Decoding coordinates with ridge regularization.** Optimally choosing the ridge regularization in the coordinate decoding task avoids the double-descent peak in test error. For convenience, we reproduce the right panel of Figure 2 on the right.

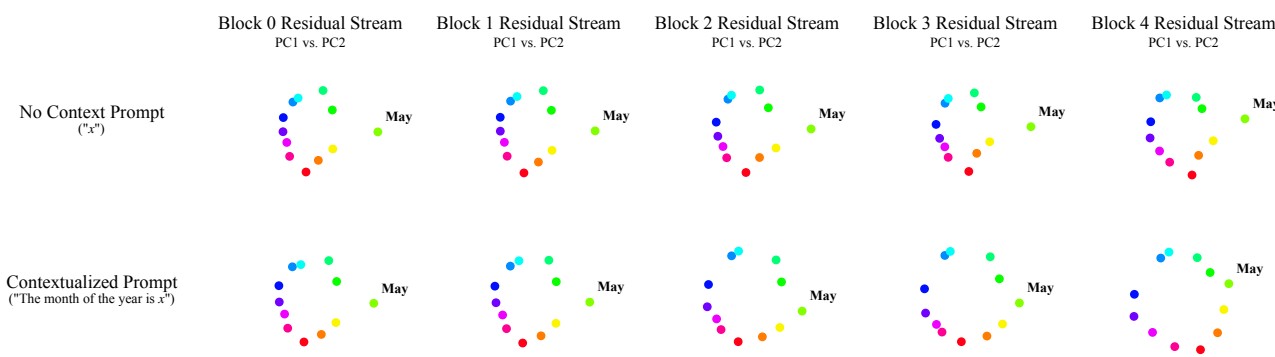

*Figure 14.* **Context allows polysemantic ambiguity to be resolved by the LLM.** Prompted with single token months, the embedding for May is distorted relative to the circle and remains that way throughout the layers of the LLM. After adding context to the prompts: "The month of the year is $x$" for $x \in [\text{January}, \ldots, \text{December}]$, we observe that the distortion in the circular geometry associated to May fades.

### C.1. Helper-based reconstruction

Here we provide additional data supporting the helper-based reconstruction. In Figure 15, we show the reconstruction task PMI (as described in A.1.4) using non-seasonal "number" words ranging from "one" to "seventeen," which do not exhibit seasonal modulation. The reconstructed PMI and embeddings from this task show that the months are not correctly ordered, consistent with the notion that the numbers do not vary in seasonality and therefore fail to assist in reconstruction.

We can identify the affinity of each word in the vocabulary with the month embeddings as a proxy for their seasonality. We define the month-word affinity as $\boldsymbol{A} = \boldsymbol{Q}_{\text{words}} \boldsymbol{Q}_{\text{months}}^{\dagger} \in \mathbb{R}^{V \times 12}$ where $\boldsymbol{Q}_{\text{months}} \in \mathbb{R}^{12 \times K}$ for embedding dimension $K$ and vocabulary size $V$. By weighting each month's affinity by $e^{i2\pi m/12}$ for month $m$, we can extract a seasonality score from the phase of each word's resulting complex affinity. Sorting words by the magnitude of the complex affinity (we take large magnitude to indicate a clear seasonal signal), we select the top-100 seasonal words and perform the reconstruction task again. As shown in Figure 16, these words produce a very high quality reconstruction.

In Figure 17, we show the reconstruction error versus the number of helper words $H$ used in the reconstruction task, where the helpers are chosen from the top-100 seasonal words or randomly from the vocabulary. We see that the reconstruction error decreases as we increase the number of helper words, and that the top seasonal words yield a faster scaling and lower reconstruction error than random words, with an error scaling as $1/\sqrt{H}$.

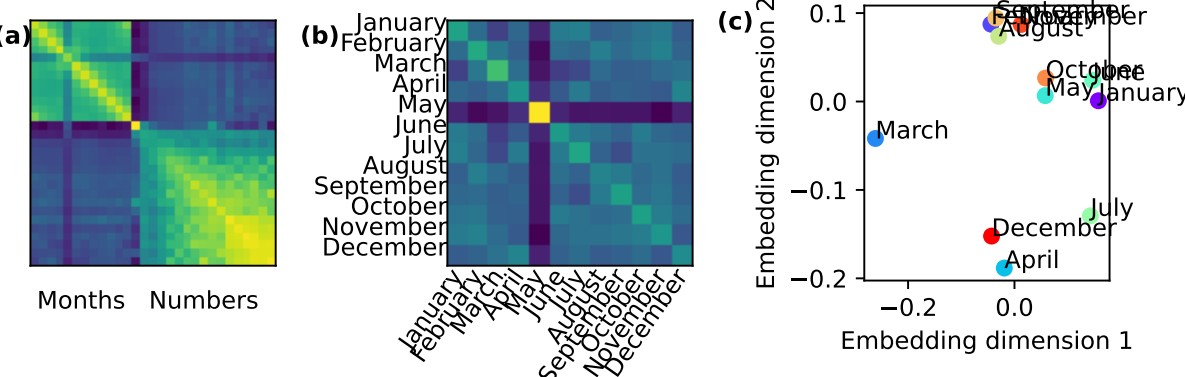

*Figure 15.* **Number words are non-seasonal and do not enable reconstruction of seasonality.** (a) Reconstruction task PMI with number words ("one", "two", . . ., "seventeen") plotted versus months of year, showing no seasonal modulation in the PMI coupling the numbers and the months. (b) Reconstructed PMI for the months, generated by the Gram matrix of the embeddings for the months derived from (a). (c) Embeddings from the reconstruction task, showing no clear ordering of the months.

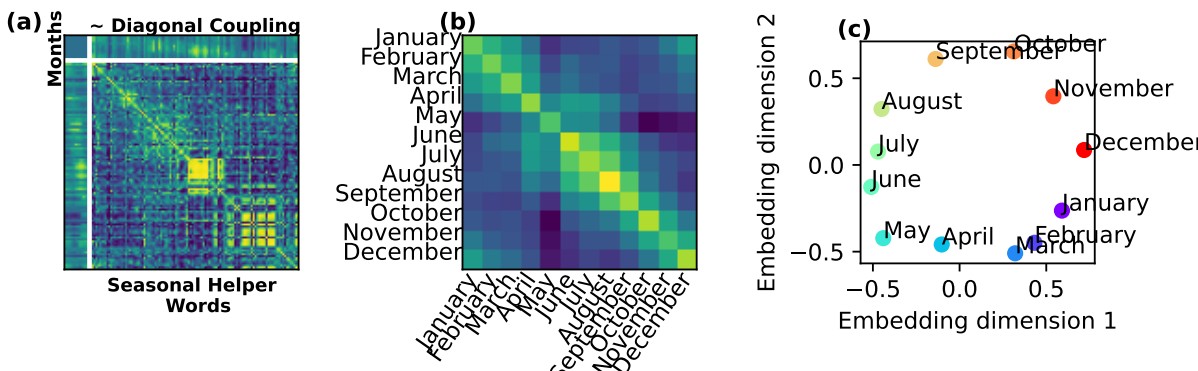

*Figure 16.* **Reconstruction with seasonal words.** (a) Reconstruction task PMI with the top-100 seasonal words plotted versus months of year, showing strong seasonal modulation. Indices of the auxilliary word subset have been ordered by their estimated seasonality score. (b) Reconstructed PMI, generated by the Gram matrix of the embeddings for the months derived from a. (c) Embeddings from the reconstruction task, showing the correct ordering of the months.

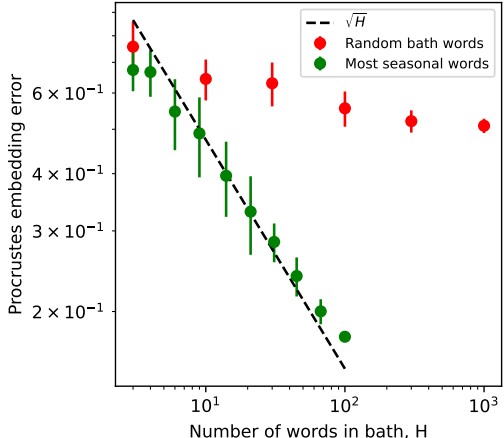

*Figure 17.* **Reconstruction error with H.** Here we plot the Procrustes error versus the number of helper words $H$ used in the reconstruction task, where the helpers are chosen from the top-100 seasonal words or randomly from the vocabulary. We see that the reconstruction error decreases as we increase the number of helper words.

## D. Combined Model of Seasonality and Binary Semantic Attributes

In Section 4.1, we consider words sharing a continuous latent attribute, but lacking any other additional semantic information. This is an unrealistic assumption, and predicts that the PCA dimensions are strictly Fourier. Here, we now enrich the vocabulary by assigning to each word both a seasonal phase $x \in \{0, \ldots, N-1\}$ and a binary attribute vector $\boldsymbol{a} = (a_1, \ldots, a_d) \in \{\pm 1\}^d$, where the binary attributes encode additional semantic information (e.g. gender, singular vs. plural, royal vs non-royal, etc...), as in Korchinski et al. (2025). The total vocabulary size is $N2^d$.

**Multiplicative generative model.** Seasonality and semantic attributes are assumed to influence co-occurrence independently. The seasonal factor is the same as in Section 4:

$$F_{\text{time}}(x, y) = 1 + C(t_x - t_y),$$

where $C$ is the circular autocorrelation of a unimodal kernel $g$.

For semantic attributes we use a fully factorized multiplicative model:

$$F_{\text{attr}}(\boldsymbol{a}, \boldsymbol{b}) = \prod_{r=1}^{d} (1 + s_r a_r b_r), \tag{42}$$

where $s_r$ measures the strength of attribute $r$.

Thus the generative probability of observing $(x, \boldsymbol{a})$ and $(y, \boldsymbol{b})$ jointly is

$$P((x, \boldsymbol{a}), (y, \boldsymbol{b})) = P(x, \boldsymbol{a}) P(y, \boldsymbol{b}) \, F_{\text{time}}(x, y) \, F_{\text{attr}}(\boldsymbol{a}, \boldsymbol{b}). \tag{43}$$

**Exact PMI and decomposition into constant and structured terms** The PMI is

$$\text{PMI}((x, \boldsymbol{a}), (y, \boldsymbol{b})) = \log F_{\text{time}}(x, y) + \log F_{\text{attr}}(\boldsymbol{a}, \boldsymbol{b}).$$

The semantic factor contributes

$$\log F_{\text{attr}}(\boldsymbol{a}, \boldsymbol{b}) = \sum_{r=1}^{d} \log(1 + s_r a_r b_r). \tag{44}$$

Because $a_r b_r \in \{\pm 1\}$, each term admits the exact decomposition

$$\log(1 + s_r a_r b_r) = \alpha_r + \beta_r \, a_r b_r,$$

with

$$\alpha_r = \tfrac{1}{2}[\log(1 + s_r) + \log(1 - s_r)], \qquad \beta_r = \tfrac{1}{2}[\log(1 + s_r) - \log(1 - s_r)].$$

Hence

$$\log F_{\text{attr}}(a, b) = A + \sum_{r=1}^{d} \beta_r \, a_r b_r, \qquad A := \sum_{r=1}^{d} \alpha_r. \tag{45}$$

The constant $A$ affects all pairs equally, whereas the second term encodes the semantic structure.

For the seasonal part we define:

$$\log F_{\text{time}}(x, y) = K_t(t_x - t_y).$$

Combining these contributions yields the *exact* PMI up to seasonal linearization:

$$\boxed{\text{PMI}((x, \boldsymbol{a}), (y, \boldsymbol{b})) = K_t(t_x - t_y) + A + \sum_{r=1}^{d} \beta_r \, a_r b_r.} \tag{E.1}$$

**Matrix structure on the product space**  Let $\boldsymbol{J}_{\text{time}}$ and $\boldsymbol{J}_{\text{attr}}$ denote the all-ones matrices of sizes $N$ and $2^d$, respectively. Define

$$K(x, y) = K_t(t_x - t_y), \qquad K_{\text{attr}}(\boldsymbol{a}, \boldsymbol{b}) = \sum_{r=1}^{d} \beta_r a_r b_r.$$

Then (E.1) takes the exact Kronecker form

$$\boxed{\mathbf{PMI} = \boldsymbol{K}_t \otimes \boldsymbol{J}_{\text{attr}} \;+\; \boldsymbol{J}_{\text{time}} \otimes \boldsymbol{K}_{\text{attr}} \;+\; A\,\boldsymbol{J}_{\text{time}} \otimes \boldsymbol{J}_{\text{attr}}.} \tag{E.2}$$

Each term acts on a distinct tensor component:

- $\boldsymbol{K}_t$ and $\boldsymbol{J}_{\text{time}}$ act on seasonal indices,

- $\boldsymbol{K}_{\text{attr}}$ and $\boldsymbol{J}_{\text{attr}}$ act on attribute indices.

**Spectral decomposition**  We diagonalize the PMI in the product basis of: (i) Fourier modes $\phi_k$ on the seasonal circle, and (ii) Walsh characters $\psi_S$, which serve as the equivalent Fourier basis on the hypercube. For any subset of attribute indices $S \subseteq \{1, 2, \ldots, d\}$, the Walsh character $\Psi_S(\boldsymbol{a})$ is defined $\Psi_S(\boldsymbol{a}) \equiv \prod_{r \in S} \boldsymbol{a}_r$.

**Seasonal operators.**  As before,

$$\boldsymbol{K}_t \phi_k = \mu_k\, \phi_k, \qquad \boldsymbol{J}_{\text{time}} \phi_0 = N\, \phi_0, \qquad \boldsymbol{J}_{\text{time}} \phi_{k \neq 0} = 0.$$

where the relation on

**Attribute operators.**  Walsh characters satisfy

$$\boldsymbol{J}_{\text{attr}} \psi_\varnothing = 2^d \psi_\varnothing, \qquad \boldsymbol{J}_{\text{attr}} \psi_{S \neq \varnothing} = 0,$$

and

$$\boldsymbol{K}_{\text{attr}} \psi_{\{r\}} = 2^d \beta_r\, \psi_{\{r\}}, \qquad \boldsymbol{K}_{\text{attr}} \psi_S = 0 \text{ if } S = \varnothing \text{ or } |S| > 1.$$

which follows from the fact that $\boldsymbol{K}(\boldsymbol{a}, \boldsymbol{b}) = \sum_r^d \beta_r \psi_{\{r\}}(\boldsymbol{a}) \psi_{\{r\}}(\boldsymbol{b})$. Note this implies $\boldsymbol{K}$ is rank $d$.

**Joint eigenbasis.**  All terms in (E.2) are diagonal in the product basis

$$\Phi_{k,S}(x, \boldsymbol{a}) = \phi_k(x) \psi_S(\boldsymbol{a}).$$

The three components contribute:

$$(\boldsymbol{K}_t \otimes J_{\text{attr}}) : \qquad 2^d\, \mu_k \text{ on } S = \varnothing;$$
$$(\boldsymbol{J}_{\text{time}} \otimes \boldsymbol{K}_{\text{attr}}) : \qquad N\, 2^d \beta_r \text{ on } (k = 0, S = \{r\});$$
$$(A\,\boldsymbol{J}_{\text{time}} \otimes \boldsymbol{J}_{\text{attr}}) : \qquad A\, N 2^d \text{ only on } (k = 0, S = \varnothing).$$

Thus:

**Theorem 5** (Spectrum of the combined seasonal–attribute PMI). *For the PMI defined in* (E.1)*, the eigenvectors are* $\Phi_{k,S}(x, a) = \phi_k(x) \psi_S(a)$ *with eigenvalues*

$$\lambda_{k,S} = \begin{cases} 2^d \mu_k + A\, N 2^d, & k = 0,\ S = \varnothing, \\ 2^d \mu_k, & k \neq 0,\ S = \varnothing, \\ N\, 2^d \beta_r, & k = 0,\ S = \{r\}, \\ 0, & \textit{otherwise.} \end{cases} \tag{2.3}$$

Crucially, the PMI decomposes into orthogonal seasonal and binary attributes subspaces. All the geometric structures obtained in the main text can thus be obtained in this model again by projecting on the space of the $\phi_k(x) \psi_{S=\varnothing}$.

# E. Proofs

## E.1. Latent semantic lattice

Let $\mathcal{S}$ be any subset of the vocabulary with cardinality $|\mathcal{S}| = L^D$, for positive integers $L$ and $D$. To define the latent semantic lattice, we first define the centered index set $\mathcal{N}$ as

$$\mathcal{N} := \begin{cases} \{-\frac{L}{2}, -\frac{L}{2} + 1, \ldots, \frac{L}{2} - 1\}^D, & L \text{ even,} \\ \{-\frac{L-1}{2}, -\frac{L-1}{2} + 1, \ldots, \frac{L-1}{2}\}^D, & L \text{ odd.} \end{cases} \tag{46}$$

This defines a simple integer lattice in $D$ dimensions with $L$ sites along each axis; thus $|\mathcal{N}| = L^D$.

For each word $i \in \mathcal{S}$, let $i \mapsto \boldsymbol{n}_i$ be a bijective map from a word to its lattice index. Define the latent semantic continuum coordinate for word $i$ as $\boldsymbol{x}_i := 2\boldsymbol{n}_i / L$. Thus the latent coordinates constitute a lattice on the interval $[-1, 1]^D$.

### E.1.1. WAVEVECTORS FOR PERIODIC BC

With periodic boundary conditions, we may define the associated reciprocal lattice. We begin with the auxiliary notion of a positive half-space: we write $\boldsymbol{n} \succ \boldsymbol{0}$ if the first nonzero component of $\boldsymbol{n}$ is positive, i.e.,

$$\boldsymbol{n} \succ \boldsymbol{0} \iff \exists j \in \{1, \ldots, D\} \text{ such that } n_1 = \cdots = n_{j-1} = 0 \text{ and } n_j > 0. \tag{47}$$

Next, for each $\boldsymbol{n} \in \mathcal{N}$ we define the periodized negation $\ominus\boldsymbol{n} \in \mathcal{N}$ to be the unique element satisfying

$$\ominus\boldsymbol{n} \equiv -\boldsymbol{n} \pmod{L} \quad \text{(componentwise).} \tag{48}$$

Then we may define the following disjoint sets:

$$\mathcal{K}_+ := \{\pi\boldsymbol{n} : \boldsymbol{n} \in \mathcal{N} \text{ and } \ominus\boldsymbol{n} \neq \boldsymbol{n} \text{ and } \boldsymbol{n} \succ \boldsymbol{0}\}, \tag{49}$$

$$\mathcal{K}_- := \{\ominus\boldsymbol{k} : \boldsymbol{k} \in \mathcal{K}_+\}, \tag{50}$$

$$\mathcal{K}_{\mathrm{sc}} := \{\pi\boldsymbol{n} : \boldsymbol{n} \in \mathcal{N} \setminus \{\boldsymbol{0}\} \text{ and } \ominus\boldsymbol{n} = \boldsymbol{n}\}, \tag{51}$$

$$\mathcal{K}_0 := \{\boldsymbol{0}\}. \tag{52}$$

Their union comprises the set of wavevectors allowed by the periodic boundary conditions:

$$\mathcal{K} := \mathcal{K}_+ \cup \mathcal{K}_- \cup \mathcal{K}_{\mathrm{sc}} \cup \mathcal{K}_0 \tag{53}$$

$$= \{\pi\boldsymbol{n}_i : i \in \mathcal{S}\}. \tag{54}$$

Despite being constructed from the same index set, the lattice coordinates $\{\boldsymbol{x}_i\}_i$ and the wavevectors $\{\boldsymbol{k}_\mu\}_\mu$ are not commensurate. The coordinates occupy a lattice in real space whereas the wavevectors live on a lattice in reciprocal space.

Each wavevector $\boldsymbol{k} \in \mathcal{K}$ is associated with the plane wave mode $\exp(i\boldsymbol{k}^\top \boldsymbol{x})$. $\mathcal{K}_+$ and $\mathcal{K}_-$ correspond to $\pm$ mode pairs, $\mathcal{K}_{\mathrm{sc}}$ corresponds to nonzero self-conjugate modes, and $\mathcal{K}_0$ corresponds to the constant (DC) mode. When $L$ is odd, the only self-conjugate wavevector is $\boldsymbol{k} = \boldsymbol{0}$, so $\mathcal{K}_{\mathrm{sc}} = \varnothing$; when $L$ is even there are $2^D - 1$ nonzero self-conjugate modes, which lie on the faces of the Brillouin zone and correspond to plane waves at the Nyquist frequency of the lattice.

Define $P := |\mathcal{K}_+|$ and $S := |\mathcal{K}_{\mathrm{sc}}|$. We choose an ordering $\mu \mapsto \boldsymbol{k}_\mu$ of the wavevectors $\mathcal{K}$ such that

$$\mathcal{K} = \{\boldsymbol{k}_{2p-1}, \boldsymbol{k}_{2p}\}_{p=1}^P \cup \{\boldsymbol{k}_{2P+s}\}_{s=1}^S \cup \{\boldsymbol{k}_{2P+S+1}\}, \tag{55}$$

with

$$\boldsymbol{k}_{2p-1} \in \mathcal{K}_+, \qquad \boldsymbol{k}_{2p} = \ominus\boldsymbol{k}_{2p-1} \in \mathcal{K}_-, \qquad \boldsymbol{k}_{2P+s} \in \mathcal{K}_{\mathrm{sc}}, \qquad \boldsymbol{k}_{2P+S+1} = \boldsymbol{0}. \tag{56}$$

Thus $\mathcal{K}$ contains first the conjugate pairs (in arbitrary order), followed by the nonzero self-conjugate modes (arbitrary order), followed by the zero mode.

### E.2. Proof of Fourier representation geometry (periodic lattice)

**Proposition 1, precise statement.** Let $\mathcal{S}$ be any subset of the vocabulary. Using the notation defined in Appendix E.1, we define the associated latent semantic lattice $\{x_i\}$ as well as the associated allowed wavevectors $\mathcal{K}$ with enumeration $\{k_\mu\}$ and partition indices $P, S$. Let $M^\star \in \mathbb{R}^{V \times V}$ denote the normalized co-occurrence matrix.

Assume Assumption B.1 holds on $\mathcal{S}$ with co-occurrence convolution kernel $C(\cdot)$. Define $\tilde{m}(k) := \mathcal{F}_x[C(\mathrm{dist}(x, 0))]$, the Fourier-domain transfer function associated with the lattice impulse response $m(x) := C(\mathrm{dist}(x, 0))$.

Let $W$ denote the word embeddings obtained as specified in Appendix B.2. Assume the embedding dimension satisfies $d \geq \mathrm{rank}\, M^\star$. Let $\overline{W}_\mathcal{S}$ denote the PCA projection of the embeddings for $\mathcal{S}$ as specified in Section 2.1.

Then, up to orthogonal transformations within degenerate subspaces, and up to permutations of the principal directions, the PCA-aligned embeddings are given by

$$
\overline{W}_{\mathcal{S}(i\mu)} = \begin{cases} \sqrt{\dfrac{2|\tilde{m}(k_\mu)|}{|\mathcal{S}|}} \sin(k_\mu^\top x_i) & \text{if } \mu \leq 2P \text{ and } \mu \text{ is odd,} \\[2ex] \sqrt{\dfrac{2|\tilde{m}(k_\mu)|}{|\mathcal{S}|}} \cos(k_\mu^\top x_i) & \text{if } \mu \leq 2P \text{ and } \mu \text{ is even,} \\[2ex] \sqrt{\dfrac{|\tilde{m}(k_\mu)|}{|\mathcal{S}|}} \cos(k_\mu^\top x_i) & \text{if } 2P < \mu \leq (2P + S), \\[2ex] 0 & \text{otherwise.} \end{cases} \tag{57}
$$

**Proof.** Let $|M^\star|$ denote the matrix absolute value of $M^\star$. Let $H \in \mathbb{R}^{|\mathcal{S}| \times |\mathcal{S}|}$ denote the submatrix of $|M^\star|$ corresponding to words in $\mathcal{S}$. Define the mean-centering projector $P := I - |\mathcal{S}|^{-1} \mathbf{1}\mathbf{1}^\top$.

Since $W$ has no rank constraint with respect to $M^\star$, it follows from Appendix B.2 that $WW^\top = |M^\star|$. Therefore the PCA-projected embeddings $\overline{W}_\mathcal{S}$ factorize the centered submatrix of $|M^\star|$:

$$
\overline{W}_\mathcal{S}\overline{W}_\mathcal{S}^\top = PHP^\top \tag{58}
$$

and the PCA geometry of the projected embeddings can be directly obtained from the spectral decomposition of $PHP^\top$.

In general, knowing the spectral decomposition of $H$ does not allow one to directly infer the spectral decomposition of its centered version. However, in this case, we will find that the constant mode $\mathbf{1}$ is an eigenvector of $H$, so centering simply removes this mode without mixing eigenvectors, allowing the remaining spectrum to be read off directly. With this knowledge, we may begin by explicitly diagonalizing $H$.

We will follow the standard argument for diagonalizing block-circulant matrices with circulant blocks. Let $v \in \mathbb{R}^{|\mathcal{S}|}$ be a candidate eigenvector of $H$. We define $v(x_i) := v_{(i)}$ and use a plane wave ansatz: $v(x_i) = \exp(\mathrm{i}k^\top x_i)$, where $\mathrm{i}$ is the imaginary unit and $k \in \mathcal{K}$. Then the $j$-th coordinate of the image of $v$ is

$$
[Hv]_{(j)} = \sum_{i=1}^{|\mathcal{S}|} C(\mathrm{dist}(x_i, x_j))\, v(x_i) \tag{59}
$$

$$
= \sum_{a \in \{x_i\}} C(\mathrm{dist}(0, a)) \exp(\mathrm{i}k^\top(x_j + a)) \tag{60}
$$

$$
= \left( \sum_{a \in \{x_i\}} C(\mathrm{dist}(0, a)) \exp(\mathrm{i}k^\top a) \right) v_{(j)} \tag{61}
$$

$$
= \tilde{m}(k)v_{(j)}, \tag{62}
$$

where in the second step we use the periodic lattice symmetry and perform the change of variables $a := x_i - x_j$, and the final step simply recognizes the discrete Fourier transform on the lattice. This confirms that $v$ is a (complex-valued) eigenvector of $H$ with eigenvalue $\tilde{m}(k)$. The eigenvalue is real, by the spectral theorem for real symmetric matrices.

The function $C(\mathrm{dist}(\mathbf{0}, \boldsymbol{x}))$ is real-valued and even in $\boldsymbol{x}$, implying conjugate symmetry in Fourier space: $\tilde{m}(\boldsymbol{k}) = \tilde{m}(\ominus \boldsymbol{k})$. Therefore the eigenvectors come in degenerate pairs, and we may superpose them in the standard way to obtain real-valued Fourier modes. These Fourier eigenvectors, considered over all lattice wavevectors $\boldsymbol{k}$, form the standard orthonormal $D$-dimensional discrete Fourier basis and hence span $\mathbb{R}^{|\mathcal{S}|}$. Thus, we have fully diagonalized $\boldsymbol{H}$:

$$
\begin{aligned}
\boldsymbol{H}_{(ij)} = \quad & \sum_{\text{odd } \mu \leq 2P} \tilde{m}(\boldsymbol{k}_\mu) \frac{\sin\left(\boldsymbol{k}_\mu^\top \boldsymbol{x}_i\right) \sin\left(\boldsymbol{k}_\mu^\top \boldsymbol{x}_j\right)}{|\mathcal{S}|/2} \\
+ & \sum_{\text{even } \mu \leq 2P} \tilde{m}(\boldsymbol{k}_\mu) \frac{\cos\left(\boldsymbol{k}_\mu^\top \boldsymbol{x}_i\right) \cos\left(\boldsymbol{k}_\mu^\top \boldsymbol{x}_j\right)}{|\mathcal{S}|/2} \\
+ & \sum_{2P < \mu \leq 2P+S} \tilde{m}(\boldsymbol{k}_\mu) \frac{\cos\left(\boldsymbol{k}_\mu^\top \boldsymbol{x}_i\right) \cos\left(\boldsymbol{k}_\mu^\top \boldsymbol{x}_j\right)}{|\mathcal{S}|} \\
+ & \tilde{m}(\mathbf{0}) \frac{1}{|\mathcal{S}|}.
\end{aligned}
\tag{63}
$$

The third sum is over nonzero self-conjugate modes; since these modes are at the Nyquist frequency, their values simply alternate $\pm 1$ on the lattice. Therefore the eigenvector normalization is $\sqrt{1/|\mathcal{S}|}$ rather than $\sqrt{2/|\mathcal{S}|}$. The same holds for the constant mode (final term).

It is easy to verify that centering annihilates the constant mode. Therefore the Gram matrix of the centered embeddings satisfy

$$
\begin{aligned}
\left[\overline{\boldsymbol{W}}_\mathcal{S} \overline{\boldsymbol{W}}_\mathcal{S}^\top\right]_{(ij)} = \quad & \sum_{\text{odd } \mu \leq 2P} \tilde{m}(\boldsymbol{k}_\mu) \frac{\sin\left(\boldsymbol{k}_\mu^\top \boldsymbol{x}_i\right) \sin\left(\boldsymbol{k}_\mu^\top \boldsymbol{x}_j\right)}{|\mathcal{S}|/2} \\
+ & \sum_{\text{even } \mu \leq 2P} \tilde{m}(\boldsymbol{k}_\mu) \frac{\cos\left(\boldsymbol{k}_\mu^\top \boldsymbol{x}_i\right) \cos\left(\boldsymbol{k}_\mu^\top \boldsymbol{x}_j\right)}{|\mathcal{S}|/2} \\
+ & \sum_{2P < \mu \leq 2P+S} \tilde{m}(\boldsymbol{k}_\mu) \frac{\cos\left(\boldsymbol{k}_\mu^\top \boldsymbol{x}_i\right) \cos\left(\boldsymbol{k}_\mu^\top \boldsymbol{x}_j\right)}{|\mathcal{S}|}
\end{aligned}
\tag{64}
$$

Since the sinusoidal modes are mutually orthogonal, the Gram matrix is already expressed in the diagonalized form $\overline{\boldsymbol{W}}_\mathcal{S} \overline{\boldsymbol{W}}_\mathcal{S}^\top = \boldsymbol{\Phi} \boldsymbol{\Lambda} \boldsymbol{\Phi}^\top$, where the columns $\boldsymbol{\Phi}_{(\cdot \mu)}$ encode the normalized sinusoids, $\boldsymbol{\Lambda}_{(\mu\mu)} = \tilde{m}(\boldsymbol{k}_\mu)$, and $\boldsymbol{D} := \mathrm{sign}(\boldsymbol{\Lambda})$. Recalling the definition of PCA projection given in Section 2.1, we recognize that this immediately yields PCA coordinates of $\boldsymbol{W}_\mathcal{S}$. Proposition 1 follows immediately.

∎

### E.3. Proof of Fourier representation geometry under exponential kernel (periodic BC)

**Corollary 2, precise statement.** Let $\mathcal{S}$ be any subset of the vocabulary. Using the notation defined in Appendix E.1, we define the associated $1D$ latent semantic lattice $\{x_i\}$ with length $L$, as well as the associated allowed wavenumbers $\mathcal{K}$ with enumeration $\{k_\mu\}$ and partition indices $P, S$. Let $M^\star \in \mathbb{R}^{\bar{V} \times V}$ denote the normalized co-occurrence matrix. Let $\overline{W}_{\mathcal{S}}$ denote the PCA-projected embeddings for $\mathcal{S}$ as specified in Appendix B.2 and Section 2.1.

Assume Assumption B.1 holds on $\mathcal{S}$. Assume the co-occurrence convolution kernel has the functional form $C(\Delta x) = \sum_{n \in \mathbb{Z}} \exp(-|\Delta x + 2n|/\sigma)$, where $\sigma$ is a free parameter. Assume the embedding dimension satisfies $d \geq \text{rank } M^\star$.

Then, up to orthogonal transformations within degenerate subspaces, and up to permutations of the principal directions, the PCA-projected embeddings are given by

$$
\overline{W}_{\mathcal{S}(i\mu)} = \begin{cases}
\sqrt{\frac{2}{L}} a_\mu \sin(k_\mu x_i) & \text{if } \mu \leq 2P \text{ and } \mu \text{ is odd,} \\[2ex]
\sqrt{\frac{2}{L}} a_\mu \cos(k_\mu x_i) & \text{if } \mu \leq 2P \text{ and } \mu \text{ is even,} \\[2ex]
\sqrt{\frac{1}{L}} a_\mu \cos(k_\mu x_i) & \text{if } 2P < \mu \leq (2P + S), \\[2ex]
0 & \text{otherwise,}
\end{cases}
\tag{65}
$$

where

$$
a_\mu = \sqrt{\frac{2}{L} \frac{1 - q^2}{1 - 2q \cos(2k_\mu/L) + q^2}}, \qquad q := e^{-2/(\sigma L)}
\tag{66}
$$

which has the continuum limit

$$
\lim_{L \to \infty} a_\mu = \sqrt{\frac{2\sigma}{1 + \sigma^2 k_\mu^2}}.
\tag{67}
$$

**Proof.** Applying Proposition 1, we see that Corollary 2 holds if we equate $a_\mu = \sqrt{|\tilde{m}(k_\mu)|}$. We define for convenience $q := e^{-2/(\sigma L)}$. Under the periodized exponential kernel, the transfer function is

$$
\tilde{m}(k) = \frac{2}{L} \sum_{x \in \{x_i\}} \left( \sum_{n \in \mathbb{Z}} \exp\left(-\frac{|x + 2n|}{\sigma}\right) \right) e^{-\mathrm{i}kx}
\tag{68}
$$

$$
= \frac{2}{L} \left( 1 + 2 \sum_{r \geq 1} q^r \cos(2kr/L) \right)
\tag{69}
$$

using a coordinate relabeling and standard trigonometric identities. Now, using the fact $|q| < 1$, we apply the standard geometric series identity $\sum_{r \geq 1} q^r e^{\mathrm{i}\theta r} = qe^{\mathrm{i}\theta}/(1 - qe^{\mathrm{i}\theta})$, take the real part, and obtain

$$
\tilde{m}(k) = \frac{2}{L} \left( 1 + 2 \operatorname{Re}\left( \frac{qe^{2\mathrm{i}k/L}}{1 - qe^{2\mathrm{i}k/L}} \right) \right)
\tag{70}
$$

$$
= \frac{2}{L} \frac{1 - q^2}{1 - 2q \cos(2k/L) + q^2}.
\tag{71}
$$

The $L \to \infty$ limit can be found by direct computation.

∎

**E.4. Proof of Fourier representation geometry under exponential kernel (open BC)**

**Proposition 3, precise statement.** Let $\mathcal{S}$ be any subset of the vocabulary. Using the notation defined in Appendix E.1, we define the associated $1D$ latent semantic lattice $\{x_i\}$ with length $L$ with open boundary conditions. Let $\boldsymbol{M}^\star \in \mathbb{R}^{V \times V}$ denote the normalized co-occurrence matrix. Let $\overline{\boldsymbol{W}}_{\mathcal{S}}$ denote the PCA-projected embeddings for $\mathcal{S}$ as specified in Appendix B.2 and Section 2.1.

Assume Assumption 3.1 holds on $\mathcal{S}$. Assume the co-occurrence convolution kernel has the functional form $C(\Delta x) = \exp(-|\Delta x|/\sigma)$, where $\sigma$ is a free parameter. Assume the embedding dimension satisfies $d \geq \operatorname{rank} \boldsymbol{M}^\star$.

Then in the continuum limit the PCA-aligned embeddings are given by

$$
\lim_{L \to \infty} \sqrt{L} \; \overline{\boldsymbol{W}}_{\mathcal{S}(i\mu)} = \begin{cases} a_\mu \dfrac{\sin(k_\mu x_i)}{N_\mu} & \text{if } \mu \text{ is odd,} \\[2ex] a_\mu \dfrac{\cos(k_\mu x_i) - \frac{\sin k_\mu}{k_\mu}}{N_\mu} & \text{if } \mu \text{ is even,} \end{cases}
\tag{72}
$$

where for each mode index $\mu$, the wavenumber $k_\mu > 0$, the normalization $N_\mu > 0$, and the singular value $a_\mu > 0$ are given as follows. $k_\mu$ is the unique solution to the self-consistent quantization condition

$$
k_\mu = \begin{cases} \dfrac{(\mu+1)\pi}{2} - \tan^{-1}(\sigma k_\mu), & \mu \text{ odd,} \\[2ex] \dfrac{\mu\pi}{2} + \tan^{-1}\left( \dfrac{k_\mu}{1 + \sigma(1+\sigma)k_\mu^2} \right), & \mu \text{ even.} \end{cases}
\tag{73}
$$

It follows that $k_\mu < k_{\mu+1}$. The normalization is

$$
N_\mu = \begin{cases} \sqrt{\dfrac{1}{2} - \dfrac{\sin(2k_\mu)}{4k_\mu}} & \mu \text{ odd,} \\[2ex] \sqrt{\dfrac{1}{2} + \dfrac{\sin(2k_\mu)}{4k_\mu} - \left(\dfrac{\sin k_\mu}{k_\mu}\right)^2} & \mu \text{ even.} \end{cases}
\tag{74}
$$

The singular values are

$$
a_\mu = \sqrt{\dfrac{2\sigma}{1 + \sigma^2 k_\mu^2}}.
\tag{75}
$$

**Proof.** In the continuum limit, we update our variables as follows. The lattice $\{x_i\}$ becomes the continuum $x \in [-1, 1]$ with uniform measure. Vectors become functions, and in particular, eigenmodes $\phi$ become eigenfunctions denoted $\phi(x)$. The submatrix of $|\boldsymbol{M}^\star|$ corresponding to words in $\mathcal{S}$ becomes the kernel operator denoted $H$, whose action on a function $f$ can be expressed as the integral transform

$$
(Hf)(x) := \int_{-1}^{1} C(|x - y|) \, f(y) \, dy,
\tag{76}
$$

where $C(|x - y|) = e^{-|x-y|/\sigma}$ is the co-occurrence kernel. The mean-centering matrix $\boldsymbol{P}$ becomes the self-adjoint operator $P$ that projects out the constant function:

$$
(Pf)(x) := ((I - \Pi)f)(x) = f(x) - \frac{1}{2}\int_{-1}^{1} f(y) \, dy.
\tag{77}
$$

where $I$ is identity and $\Pi$ is the rank-one projector onto the constant mode.

**Warm-up.** To obtain the embedding geometry, we must diagonalize the operator $PHP$. As a warm-up, let us first diagonalize the uncentered co-occurrence operator $H$. A general solution strategy is given by Youla (1957); here we specialize and simplify the solution for our setup. We begin by obtaining endpoint identities for functions in the image of $H$, e.g., $u = Hf$, by differentiating the integral representation of $H$ and evaluating at the endpoints:

$$
u'(1) = -\frac{1}{\sigma}u(1), \qquad u'(-1) = \frac{1}{\sigma}u(-1).
\tag{78}
$$

We then notice that $C$ satisfies the identity

$$\left(1 - \sigma^2 \partial_x^2\right) e^{-|x|/\sigma} = 2\sigma \, \delta(x). \tag{79}$$

This convenient identity states that $H$ is the Green's function of the linear differential operator

$$L := \frac{1 - \sigma^2 \partial_x^2}{2\sigma} \tag{80}$$

on $[-1, 1]$ with the boundary conditions given in Equation (78). Thus, diagonalizing $H = L^{-1}$ amounts to diagonalizing $L$, i.e., if $H\phi = \lambda\phi$ then $L\phi = \lambda^{-1}\phi$. We begin by evaluating the latter equation, yielding

$$\phi'' + \frac{1}{\sigma^2}\left(\frac{2\sigma}{\lambda} - 1\right)\phi = 0. \tag{81}$$

Associating $k^2 := \frac{1}{\sigma^2}\left(\frac{2\sigma}{\lambda} - 1\right)$, we see that this is simply the homogeneous Helmholtz equation, i.e., the harmonic oscillator ODE. This implies that the eigenfunctions must be sinusoids with wavenumber $k$. Solving $\lambda$ in terms of $k$, we find that the eigenvalues of the exponential kernel $H$ are

$$\lambda = \frac{2\sigma}{1 + \sigma^2 k^2}. \tag{82}$$

Since $H$ and its domain have parity symmetry around the origin, if $\phi(x)$ is an eigenfunction then so must be $\phi(-x)$ with the same eigenvalue. Furthermore, since $L$ defines a regular Sturm–Liouville problem on a finite interval with separated boundary conditions, we may use the known fact that such operators have non-degenerate spectra. This implies that $\phi(x) = \pm\phi(-x)$. This constraint precludes both complex plane wave solutions and sinusoids with arbitrary phase shifts; each eigenfunction must be either $\sin(kx)$ or $\cos(kx)$. Plugging these into the boundary conditions (Equation (78)) yields constraints on the possible values of $k$.

**With mean-centering.**   We now aim to obtain the spectral decomposition of the mean-centered operator $H_c := PHP$. We first observe that since $Pf = f$ for any odd function $f$, the odd sector solutions obtained above are unaffected. Let us now consider how the even modes change under mean-centering.

First we observe that since $P$ projects out the DC mode, $H_c\phi$ must have no DC component. If $\phi$ is an eigenfunction, then $\phi$ can have no DC component, and $P\phi = \phi$. Therefore the eigenproblem for $H_c$ becomes

$$\lambda\phi = PHP\phi \tag{83}$$
$$= PH\phi \tag{84}$$
$$= H\phi - \Pi H\phi. \tag{85}$$

Note that the last term is a constant function (whose amplitude depends on $\phi$ and $\lambda$). Denoting $\alpha := \Pi H\phi$, we obtain the integral equation

$$H\phi = \lambda\phi + \alpha. \tag{86}$$

Acting on both sides with $L$, we obtain

$$LH\phi = \lambda L\phi + L\alpha \tag{87}$$
$$\phi = \lambda\left(\frac{\phi - \sigma^2\phi''}{2\sigma}\right) + \frac{\alpha}{2\sigma} \tag{88}$$

which simplifies to

$$\phi'' + \frac{1}{\sigma^2}\left(\frac{2\sigma}{\lambda} - 1\right)\phi = \frac{\alpha}{\lambda\sigma^2}. \tag{89}$$

This is simply the *inhomogeneous* Helmholtz equation with constant driving. Since $\phi = \text{const.}$ is a particular solution, and since the Helmholtz equation is a linear ODE, the general solutions must be sinusoids offset by constants. We have already seen that the constant offset for the odd solutions is zero (since there we have $\alpha = 0$). Additionally, we have already

discussed that the constant offset must be exactly such that the DC component of $\phi$ vanishes. Therefore the even solutions take the form

$$\phi(x) = \cos(kx) - \frac{1}{2} \int_{-1}^{1} \cos(ky) \, dy \tag{90}$$

$$= \cos(kx) - \frac{\sin k}{k} \tag{91}$$

where, just as before, $k^2 = \frac{1}{\sigma^2} \left( \frac{2\sigma}{\lambda} - 1 \right)$. Therefore the eigenvalue $\lambda$ is unaffected by centering.

Now we impose the boundary conditions to constrain the possible values of $k$. Differentiating (86) and evaluating at $x = \pm 1$ yields the centered boundary conditions

$$\phi'(1) = -\frac{1}{\sigma} \phi(1) - \frac{\alpha}{\lambda \sigma}, \qquad \phi'(-1) = +\frac{1}{\sigma} \phi(-1) + \frac{\alpha}{\lambda \sigma}. \tag{92}$$

In the odd sector, $\alpha = 0$ and the boundary conditions reduce to those in the warm-up. Plugging in the candidate eigenfunction $\phi(x) = \sin(kx)$, we find the quantization condition

$$k \cos k = -\frac{1}{\sigma} \sin k \iff \tan k = -\sigma k. \tag{93}$$

In the even sector, we plug Equation (91) into the inhomogeneous Helmholtz equation and obtain

$$\frac{\alpha}{\lambda \sigma^2} = -k \sin k. \tag{94}$$

Evaluating the boundary conditions Equation (92) yields the quantization condition

$$\cos k = \sin k \left( \frac{1}{k} + \sigma(1 + \sigma)k \right) \iff \tan k = \frac{k}{1 + \sigma(1 + \sigma)k^2}. \tag{95}$$

Combining the odd-sector quantization (93) with the centered-even quantization (95), we may write the self-consistent enumerated forms (with $k_\mu > 0$ the unique solution in the appropriate interval for each $\mu$):

$$k_\mu = \begin{cases} \dfrac{(\mu+1)\pi}{2} - \tan^{-1}(\sigma k_\mu), & \mu \text{ odd}, \\[2ex] \dfrac{\mu\pi}{2} + \tan^{-1}\left( \dfrac{k_\mu}{1 + \sigma(1+\sigma)k_\mu^2} \right), & \mu \text{ even}. \end{cases} \tag{96}$$

Since the argument of each $\tan^{-1}(\cdot)$ is strictly positive, those terms are strictly in the interval $(0, \frac{\pi}{2})$. Therefore $k_\mu < k_{\mu+1}$.

Finally, we compute the normalizing constants for the eigenfunctions. For odd modes $\phi_\mu(x) = \sin(k_\mu x)$,

$$N_\mu^2 = \frac{1}{2} \int_{-1}^{1} \sin^2(k_\mu x) \, dx = \frac{1}{2} - \frac{\sin(2k_\mu)}{4k_\mu}. \tag{97}$$

For even modes $\phi_\mu(x) = \cos(k_\mu x) - \sin k_\mu / k_\mu$,

$$N_\mu^2 = \frac{1}{2} + \frac{\sin(2k_\mu)}{4k_\mu} - \left( \frac{\sin k_\mu}{k_\mu} \right)^2. \tag{98}$$

Since the centered operator $H_c$ is diagonalized by the above odd and centered-even eigenfunctions, with eigenvalues $\lambda_\mu = 2\sigma/(1 + \sigma^2 k_\mu^2)$, the coefficients of the word embeddings are simply the singular values $a_\mu = \sqrt{\lambda_\mu}$. This yields the stated PCA-aligned embeddings.

■

### E.5. Proof of lattice coordinate decoding

**Proposition 4, precise statement.** We work in the setting of Proposition 1, which defines the vocabulary subset $\mathcal{S}$; the lattice coordinates $\{x_i\}$ and lattice length $L$; the wavevectors $\{k_\mu\}$ and their partition indices $P, S$; the eigenvalues $\{\lambda_\mu\}$; and the PCA projection of the embeddings $\overline{W}_\mathcal{S} \in \mathbb{R}^{|\mathcal{S}| \times |\mathcal{S}|}$.

Assume $\lambda_\mu$ is monotone non-increasing with $k_\mu$. For simplicity, we assume $L$ is odd; the analysis and result is almost identical for the even case.

Let $r \leq |\mathcal{S}|$ be the embedding rank, and define $W_r$ to be the Eckart-Young-Mirsky rank-$r$ approximation of $\overline{W}_\mathcal{S}$. We assume this approximation is unique, i.e., $r$ is chosen such that there is a spectral gap at the truncation rank. (In other words, $r$ is chosen so that the truncation does not select only a strict subset of a degenerate singular subspace.)

Let $X \in \mathbb{R}^{|\mathcal{S}| \times D}$ be the matrix of lattice coordinates. Let $\Omega \in \mathbb{R}^{r \times D}$ be a linear probe. Define the decoding error as

$$\varepsilon^2(\Omega; r) := \frac{\|W\Omega - X\|_F^2}{\|X\|_F^2}. \tag{99}$$

Then

$$\min_{\Omega} \varepsilon^2(\Omega; r) \leq \frac{6}{\pi^2} \frac{L^2}{L^2 - 1} \left( \left( \frac{r}{\mathrm{Vol}_D} \right)^{1/D} - \frac{\sqrt{D}}{2} \right)^{-1}. \tag{100}$$

**Proof.** We recall from Proposition 1 that $\overline{W}_\mathcal{S} = FA$, where the eigenmodes $F_{(\cdot, \mu)}$ are normalized, discrete, real-valued Fourier modes, and $A$ is the diagonal matrix of amplitudes whose elements are given by the square root of the eigenvalues of $|M^\star|_\mathcal{S}$. It follows that $W_r = F_r A_r$.

The mean-squared-error optimal $\Omega$ is simply the OLS solution, $\hat{\Omega} = (W_r^\top W_r)^{-1} W_r^\top X$. Therefore

$$W_r \hat{\Omega} = W_r (W_r^\top W_r)^{-1} W_r^\top X \tag{101}$$

$$= F_r A_r \left( A_r F_r^\top F_r A_r \right)^{-1} A_r F_r^\top X \tag{102}$$

$$= F_r F_r^\top X \tag{103}$$

where in the last step we use the semi-orthogonality of $F_r$. Note that $F_r F_r^\top$ is an orthogonal projection matrix; it projects $X$ into its top-$r$ Fourier approximation. The remainder of the proof is in simply computing the resulting error.

Let $k_n$ denote the $n$-th smallest wavevector (by magnitude), and let $\phi_n \in \mathbb{R}^{|\mathcal{S}|}$ be the corresponding lattice Fourier mode. The ordering of wavevectors sharing the same magnitude may be arbitrary. Additionally, since $\varepsilon^2$ is invariant to rescaling of the coordinates, let us rescale $X \mapsto (L/2)X$ so that the lattice coordinates are integers. The allowed wavevectors are rescaled commensurately: $k \mapsto (2/L)k$.

Then the numerator of the minimal error is

$$\|W_r \hat{\Omega} - X\|_F^2 = \|(I_{|\mathcal{S}|} - F_r F_r^\top)X\|_F^2 \tag{104}$$

$$= \sum_{\ell=1}^{D} \sum_{n=r+1}^{|\mathcal{S}|} \left( \phi_n^\top X_{(\cdot \ell)} \right)^2. \tag{105}$$

This equation states that the coordinate decoding error decomposes into overlaps between the lattice coordinates and the short-wavelength Fourier modes not realized in the embedding vectors. To make further progress, let us unroll one of the overlap terms. WLOG let us examine the case $\ell = 1$:

$$\phi_n^\top X_{(\cdot, 1)} = \sum_{x_1=-M}^{M} x_1 \sum_{x_2=-M}^{M} \cdots \sum_{x_D=-M}^{M} \sqrt{\frac{2}{|\mathcal{S}|}} \exp \left( \mathrm{i} \sum_{p=1}^{D} (k_n)_{(p)} x_p \right) \tag{106}$$

where we use the fact the lattice length is odd, $L = 2M + 1$. We implicitly understand that this equation takes the real or imaginary parts as prescribed by $\phi_n$.

One may verify that this inner product vanishes if, for any $p \neq \ell$, the corresponding wavevector coordinate $(\boldsymbol{k}_n)_{(p)} \neq 0$. Thus, the only inner products that survive are the ones for which the wavevector is perfectly aligned with the lattice coordinate in question, i.e., $\hat{\boldsymbol{e}}_\ell^\top \boldsymbol{k}_n = |\boldsymbol{k}_n|$. Furthermore, the real part also vanishes since $f(x_\ell) = x_\ell$ is odd while $\cos(kx_\ell)$ is even. Together, this gives

$$\boldsymbol{\phi}_n{}^\top \boldsymbol{X}_{(\cdot \ell)} = \sqrt{\frac{2}{|\mathcal{S}|}} \sum_{x_1=-M}^{M} x_1 \sin(|\boldsymbol{k}_n|x_1) \sum_{x_2=-M}^{M} \cdots \sum_{x_D=-M}^{M} 1 \tag{107}$$

$$= L^{D-1} \sqrt{\frac{2}{L^D}} \sum_{x_1=-M}^{M} x_1 \sin(|\boldsymbol{k}|x_1) \tag{108}$$

if $\hat{\boldsymbol{e}}_\ell^\top \boldsymbol{k}_n = |\boldsymbol{k}_n|$ and $\boldsymbol{\phi}_n$ odd, and 0 otherwise. Therefore the squared overlap is

$$\left( \boldsymbol{\phi}_n{}^\top \boldsymbol{X}_{(\cdot \ell)} \right)^2 = \frac{2L^{D-1}}{L} \left( \sum_{x=-M}^{M} x \sin(|\boldsymbol{k}_n|x) \right)^2 . \tag{109}$$

Note that this does not depend on $\ell$. Thus, the $D$-dimensional lattice problem decouples into independent 1D problems:

$$\|\boldsymbol{W}_r \hat{\boldsymbol{\Omega}} - \boldsymbol{X}\|_F^2 = \sum_{\ell=1}^{D} \sum_{n=r+1}^{|\mathcal{S}|} \left( \boldsymbol{\phi}_n{}^\top \boldsymbol{X}_\ell \right)^2 \tag{110}$$

$$= DL^{D-1} \sum_{n=R+1}^{M} \frac{2}{L} \left( \sum_{x=-M}^{M} x \sin(2\pi nx/L) \right)^2 \tag{111}$$

where we define $R$ such that $2\pi R/L$ is the maximal wavenumber magnitude encoded in the rank $r$ embeddings. Our spectral gap condition guarantees that $R$ is well-defined. We will later find a bound for $R$ in terms of $r$.

We now evaluate the trigonometric sum:

$$\sum_{x=-M}^{M} x \sin(2\pi nx/L) = 2 \sum_{x=1}^{M} x \sin(2\pi nx/L) \tag{112}$$

$$= \frac{1}{2} \left( \frac{\sin(\pi n(L+1)/L)}{\sin^2(\pi n/L)} - \frac{(L+1)\cos(\pi n)}{\sin(\pi n/L)} \right) \tag{113}$$

$$= (-1)^{n+1} \frac{L}{2\sin(\pi n/L)} \tag{114}$$

where the sum formula can be found in Gradshteyn & Ryzhik (2014) Eq. 1.352.1 and the final step follows from a few lines of algebra. Thus the numerator simplifies further:

$$\|\boldsymbol{W}_r \hat{\boldsymbol{\Omega}} - \boldsymbol{X}\|_F^2 = DL^{D-1} \sum_{n=R+1}^{M} \frac{L}{2} \csc^2(\pi n/L). \tag{115}$$

It is now useful to compute the denominator of the error. Using a very similar decoupling argument, we find

$$\|\boldsymbol{X}\|_F^2 = \sum_{\ell=1}^{D} L^{D-1} \sum_{x=-M}^{M} x^2 \tag{116}$$

$$= DL^{D-1} \cdot 2 \cdot \frac{M(M+1)(2M+1)}{6} \tag{117}$$

$$= DL^{D-1} \frac{L(L^2-1)}{12} \tag{118}$$

using a well-known formula for the sum of a sequence of squares. Combining with the numerator, we can now obtain an expression for the relative error:

$$\min_{\boldsymbol{\Omega}} \varepsilon^2(\boldsymbol{\Omega}; r) = \frac{6}{L^2 - 1} \sum_{n=R+1}^{M} \csc^2(\pi n/L). \tag{119}$$

Since $\csc$ is positive and decreasing on this interval, we may bound the sum as

$$\sum_{n=R+1}^{M} \csc^2(\pi n/L) \leq \int_{R}^{M} \mathrm{d}t \, \csc^2(\pi t/L) \tag{120}$$

$$\leq \frac{L}{\pi} \cot \frac{\pi R}{L} - \cot \frac{\pi M}{L} \tag{121}$$

$$\leq \frac{L}{\pi} \cot \frac{\pi R}{L} \tag{122}$$

$$\leq \frac{L^2}{\pi^2 R} \tag{123}$$

$$\tag{124}$$

where the fourth step follows from $\cot(x) \leq 1/x$ on this interval. This gives an upper bound in terms of $R$:

$$\min_{\boldsymbol{\Omega}} \varepsilon^2(\boldsymbol{\Omega}; r) \leq \frac{6}{\pi^2} \frac{L^2}{L^2 - 1} \frac{1}{R}. \tag{125}$$

Now we must relate $1/R$ to the embedding rank $r$. Since the wavenumbers are monotonic decreasing with the eigenvalues, the top $r$ embedding directions contain the Fourier modes with the $r$ shortest wavevectors. Therefore, in the so-called *reciprocal lattice*, we identify $r$ as the number of integer lattice sites enclosed by a $D$-sphere of radius $R$. Using the standard volume argument for counting integer points, we recognize that

$$r \leq \mathrm{Vol}_D \cdot \left( R + \frac{\sqrt{D}}{2} \right)^D \tag{126}$$

where $\mathrm{Vol}_D$ is the volume of the unit $D$-sphere. Rearranging and substituting, we obtain

$$\varepsilon^2 \leq \frac{6}{\pi^2} \frac{L^2}{L^2 - 1} \left( \left( \frac{r}{\mathrm{Vol}_D} \right)^{1/D} - \frac{\sqrt{D}}{2} \right)^{-1}. \tag{127}$$

Note that when the lattice dimension $D = 1$ we have

$$\varepsilon^2 \leq \frac{12}{\pi^2} \frac{L^2}{L^2 - 1} \left( \frac{1}{r - 1} \right) \sim \frac{1}{r} \tag{128}$$

and when $D = 2$ we have

$$\varepsilon^2 \leq \frac{6}{\pi^2} \frac{L^2}{L^2 - 1} \left( \sqrt{\frac{r}{\pi}} - \frac{1}{\sqrt{2}} \right)^{-1} \sim \frac{1}{\sqrt{r}}. \tag{129}$$

∎

