# OpenReview forum: "Symmetries in language statistics shape the geometry of model representations"
_ICML.cc/2026/Conference — ICML 2026 spotlight_

### Official Review · Reviewer_SwGu · 2026-02-28

**Soundness:** 3
**Presentation:** 3
**Significance:** 3
**Originality:** 3
**Overall Recommendation:** 5
**Confidence:** 4

**Summary:**

Using the Fourier transformation, the authors prove theoretically that translation symmetry (in terms of PMI) leads to a previously observed embedding geometry for concepts with periodic or open boundary conditions. They empirically show that word embeddings and LLM representations align with theoretical predictions. Moreover, they explain why the error in the linear coordinate decoding task scales with the probe dimension, and show that collective effects of the vocabulary can lead to the embedding geometry.

**Compliance With Llm Reviewing Policy:**

Affirmed.

**Final Justification:**

My concerns have been addressed, and I maintain my original assessment.

**Key Questions For Authors:**

- Could you provide evidence for why LLM representations encode PMI or translation symmetry?
- Could you provide more Lissajous curves in Figure 2 for concepts with periodic boundary conditions, and also for LLM representations?

**Limitations:**

The limitation is similar to the weakness mentioned above. It is unclear why this theory should apply to LLMs.

**Strengths And Weaknesses:**

The theory is very strong and explains many empirical phenomena. In particular, Figure 2 illustrates why this theory is important for a deeper understanding of the embedding geometry. For all arguments, the authors provide sufficient theoretical and empirical evidence.

One weakness is the connection of this theory to modern LLMs. It is unclear why LLM representations would encode PMI. Even if they learn a PMI matrix, it is unclear what corresponds to $W W^\top$ in LLMs.

Some typos: $p=1$ should be $i=1$ in Line 1417, and $a: = x_p - x_q$ should be $a:= x_i - x_j$ in Line 1427. Then, $a \in \\{x_i\\}$ in Line 1420 should be changed too.

---

> ### Author Rebuttal · Authors · 2026-03-30
>
> Thank you for your positive review.
>
> #### **It is unclear why LLM representations would encode PMI ...**
>
> The goal of our work is not to provide theorems that would apply to LLMs -- this is well beyond the state of the art of what can be achieved theoretically. Our goal is instead to provide a plausible explanation for the puzzling observation that learned representations in LLMs organize into simple geometric structures.  Our demonstration that:
> * (i) in simpler algorithms like word2vec (which inspired LLMs), such geometric structure must arise due to co-occurrence statistics alone
> * (ii)  LLMs and word2vec representations are very similar
>
> are strong indications that co-occurrence is key for LLMs as well.
>
> The fact that co-occurrence statistics affect LLMs representation is not surprising in itself: LLMs first learn pairwise correlations in the data, see e.g. https://arxiv.org/abs/2410.19637. This is also true after training for the first layers of LLMs for specific hierarchical models of data https://arxiv.org/abs/2406.00048.
>
> In our mind, the fundamental question that remains is how context can be used to improve representation (e.g. to distinguish the month of May and the verb), as we document in Appendix, Fig 12. We believe our work will generate activity in this direction.
>
> We will emphasize all these points to clarify further the scope of our work in the manuscript.
>
> #### **Could you provide more Lissajous curves?**
>
> Yes, here is an example with periodic boundary conditions: https://imgur.com/a/FkcFiJ6
>
> #### **Some typos: ...**
> We will fix these typos, thank you.

---

> > ### Author Rebuttal · Reviewer_SwGu · 2026-04-01
> >
> > Thank you for the rebuttal! My concerns have been addressed, and I maintain my original assessment.

---

### Official Review · Reviewer_yctU · 2026-03-13

**Soundness:** 2
**Presentation:** 2
**Significance:** 2
**Originality:** 3
**Overall Recommendation:** 4
**Confidence:** 3

**Summary:**

This paper argues that representational geometry in neural language models reflects pairwise co-occurrence statistics between words. In particular, translation symmetries and co-occurrence statistics drive the formation of three geometric structures: circles in representation space for cyclical concepts, rippled 1D manifolds for continuous sequences, and linear decoding of spatiotemporal coordinates. The authors develop a mathematical framework showing that when co-occurrence probability depends only on temporal or spatial distance between two words, models spontaneously learn Fourier representations whose amplitude and frequency can be analytically predicted. They validate this theory in word embedding models, text embedding models, and large language models, and further argue that these structures are robust to significant perturbations of the co-occurrence statistics, explaining this robustness via a continuous latent variable model.

**Compliance With Llm Reviewing Policy:**

Affirmed.

**Final Justification:**

Rebuttal has addressed my main concerns.

**Key Questions For Authors:**

- How do the empirical results change when token frequency is explicitly controlled for?

**Limitations:**

yes

**Strengths And Weaknesses:**

__Strenghts__

- The unifying theoretical framework is the paper's most valuable contribution. Deriving multiple seemingly disparate geometric observations (circles, rippled manifolds, linear coordinate decoding) from a single principle of translation symmetry is a genuinely interesting organizing move.

- The empirical observations in word embedding models are cleanly presented and provide a useful sanity-check for the theory. The connection between PMI matrix eigendecomposition and learned geometry is well-grounded in prior work showing that models like GloVe and word2vec approximate the PMI.

- The coordinate system choice, restricting to a concept subspace S with a normalized range, is justified clearly and the geometric setup is internally consistent.

__Weaknesses__

- The most significant concern is the conflation of co-occurrence statistics with raw token frequency. A substantial body of prior work has documented that token frequency in the training corpus directly shapes the geometry of learned representations [1], including outlier dimensions [2], and isotropy [3], [4] which all impact the overall structure of embedding spaces , [2]. Co-occurrence probability between two tokens is at least partly a function of their marginal frequencies, and the paper does not adequately disentangle these two factors. Any theoretical or empirical claims about translation symmetry in co-occurrence statistics must be interpreted against this baseline, and the authors do not do so. I believe the conflation between frequency and co-occurrence is not a minor issues and threatens the interpretation of nearly all the empirical results.

- The paper makes strong theoretical predictions but provides no quantitative evaluation of those predictions against empirical results. t is unclear how much of the predicted geometry is actually recovered, under what conditions the predictions break down, and what the error tolerances are. Proposition 4 is one partial exception, but it does not compensate for the broader absence of quantitative evaluation.

- Several key concepts are introduced without sufficient definition or motivation. Periodic versus open boundary conditions, for instance, are never formally defined, and the examples given are not sufficient to ground a reader unfamiliar with the distinction.

- The extension to transformer-based language models is underdeveloped. The authors acknowledge that transformers learn contextualized representations atop low-order statistics, but the empirical validation in this setting is limited and the theoretical justification for why the word-embedding theory should transfer is largely informal.

[1] Representation Degeneration Problem in Training Natural Language Generation Models. Gao et al. 2019.
[2] Outlier Dimensions that Disrupt Transformers are Driven by Frequency. Puccetti et al. 2022
[3] IsoScore: Measuring the Uniformity of Embedding Space Utilization. Rudman et al. 2022
[4] Isotropy, Clusters, and Classifiers. Mickus et al. 2024

---

> ### Author Rebuttal · Authors · 2026-03-31
>
> We thank the referee for their detailed review.
>
> #### **the conflation between frequency and co-occurrence is not a minor issue and threatens the interpretation of nearly all the empirical results.**
>
> The primary and most significant concern of the referee comes from a misunderstanding: the token frequency (unigram statistics) _are_ already controlled for in our framework. As is already apparent in Eq.1, the quantity being learned is not the raw co-occurrence probability Pr(i,j), but rather a function of the relative co-occurrence $C_{ij} := \mathrm{Pr}(i,j)/\mathrm{Pr}(i)\mathrm{Pr}(j)$. The unigram statistics are thus accounted for, automatically controlling the distortion induced by Zipf’s law. This idea has been known for a long time, see Levy and Goldberg 2014 ([Neural Word Embedding as Implicit Matrix Factorization](https://proceedings.neurips.cc/paper_files/paper/2014/file/b78666971ceae55a8e87efb7cbfd9ad4-Paper.pdf)), or even inverse document frequency from Jones 1972 ([A statistical interpretation of term specificity and its application in retrieval](https://www.staff.city.ac.uk/~sbrp622/idfpapers/ksj_orig.pdf)).
>
> It is possible that the reviewer’s misunderstanding was due to our use of the shorthand “co-occurrence” to refer to the matrix $M^*_{ij}=\log(C_{ij})$, rather than the raw probabilities $P_{ij}$. We used this term for convenience, because in fact it is the pairwise statistics (and not the unigram) that lends all the rich spectral structure: more explicitly, the unigram part alone, $C_{ij} = \mathrm{const}/(P_i P_j)$, would only be rank 1 and would play a trivial role in the geometry. We will edit the text to clarify this.
>
> #### **The paper makes strong theoretical predictions but provides no quantitative evaluation of those predictions against empirical results...**
>
> We respectfully disagree. We believe our empirical evaluations are thorough, especially compared with prior work. For example, Fig 1 shows agreement in the representational Gram matrices, which indicates that our predictions for the eigenvectors and their ordering will hold beyond the three PCA directions visualized. Figs 2 (left) and 9 show excellent pointwise agreement between multiple fixed low-d projections of the set of representations, which is very nontrivial in high dimension. Figs 2 (right) and 10 quantify the transferability of these learned representations for a downstream task. Figs 4 and 15 demonstrate the robustness of our predictions to perturbations such as sampling noise and ancillary semantic signals, following standard matrix perturbation theory. Figs 6 and 7 directly and quantitatively check the main modeling assumption of our theory.
>
> In the main figures, we chose to demonstrate our theory using PCA projections because this is how the original empirical observations were communicated in their respective papers. It was a deliberate choice to emphasize the striking visual concordance between representations, which does not reflect an absence of quantitative agreement. To emphasize the quantitative agreement as well, we will include the following new plots in the appendix: https://imgur.com/a/FkcFiJ6
>
> #### **Several key concepts are introduced without sufficient definition or motivation...**
>
> We will clarify the text to include definitions of periodic and open boundary conditions, which are standard notions in some fields but may not be in others. Systems with periodic boundary conditions loop around on themselves (for example the months of the year, or any set of integers under modular arithmetic). Open boundaries do not loop, e.g., the years of the 20th century, for which the endpoints 1900 and 1999 are not adjacent. Therefore, the boundary conditions directly influence how distances are computed on the semantic lattice. This is important because our theory connects lattice distances to spectral structure (Assumption 3.1), and then spectral structure to representational geometry (eq 2).
>
> #### **The extension to transformer-based language models is underdeveloped...**
>
> While it is an interesting open problem to study the details of how transformer-based models learn, a formal analysis of transformer training dynamics is currently well beyond the state of the field. Yet the conceptual framework we provide for how learned representations emerge leads to a plausible answer for observations that were previously very mysterious. This framework is theoretically grounded in a linear language model (word2vec), empirically validated for more complex models such as LLMs, and connects to prior work on LLM training dynamics (see our response to reviewer SwGu). We identified potential limitations of this framework, in particular the fact that transformer-based models can refine the representation geometry using contextual clues (see Fig 12).
>
> We hope that these clarifications adequately address your concerns and thus lead to an increased score. Otherwise, we would be happy to answer any remaining questions you may have.

---

> > ### Author Rebuttal · Reviewer_yctU · 2026-04-01
> >
> > My concerns have been addressed.

---

### Official Review · Reviewer_GdUL · 2026-03-13

**Soundness:** 3
**Presentation:** 4
**Significance:** 2
**Originality:** 3
**Overall Recommendation:** 5
**Confidence:** 3

**Summary:**

The paper investigates the emergence of geometric structures in neural network representations, predicting an appearance of such structures under translation symmetry in pairwise co-occurrence statistics of words, linking the effects observed in both static and modern deep contextual language models. The authors provide theoretical and empirical evidence that concepts like calendar months (circular geometry), historical years (1D manifold with ripples), and geographic locations (linear decodable coordinates) acquire specific geometric arrangements in embedding spaces. The key insight is that when co-occurrence probabilities depend only on the distance between words on an underlying latent continuum (e.g., time or space), learned embeddings naturally form Fourier modes, resulting in the observed geometric structures. The paper further shows these structures persist even under significant perturbations to co-occurrence statistics, a phenomenon the authors attribute to the collective effect of many words sharing the same continuous latent variable. Although the contributions seem somewhat incremental, the strengths outweight several limitations, so I think the paper is worth publishing.

**Compliance With Llm Reviewing Policy:**

Affirmed.

**Final Justification:**

The authors have answered all my questions, and I have no further concerns regarding the paper.

**Key Questions For Authors:**

Please see the section above.

**Limitations:**

yes

**Strengths And Weaknesses:**

Strengths:
1. The paper presents rigorous derivations connecting co-occurrence statistics to embedding geometry. The authors provide a theory behind the Fourier representation geometry for periodic and open boundary conditions, obtaining precise predictions for eigenmodes and amplitudes.

2. The work integrates various empirical observations from the literature (circular representations for cyclical concepts, rippled 1D manifolds for continuous sequences, and linear decodability of spatiotemporal coordinates) under a single principle rooted in translation symmetry.

3. The authors validate their theory across multiple model types: word embedding models (word2vec-style), text embedding models (EmbeddingGemma), and large language models (Gemma 2 2B), demonstrating broad transferability of underlying principles.

4. Aside from the theory, explaining various effects, the paper makes testable predictions, such as the existence of Lissajous curves in embedding projections (Figure 2, left) and the scaling of linear probe error with embedding dimension (Proposition 4), which are then empirically confirmed.

Weaknesses:
1. The theoretical results depend heavily on Assumption 3.1 (translation symmetry in co-occurrence statistics). While the authors show this holds approximately for real data (Figures 6 and 7), the assumptions may be too strong. However, I don't consider it a major weakness as the authors state it explicitly and provide extensive experiments regarding the topic.

2. While the paper mentions limitations regarding the extension to LLMs, it does not discuss whether such observations would be transferable or cause potential failure cases, such as in low-resourse  languages with different temporal concepts or representations arising from multimodal embeddings. Do you have any intuition regarding the topic?

---

> ### Author Rebuttal · Authors · 2026-03-30
>
> Thank you for your detailed and positive review.
>
> #### **The theoretical results depend heavily on Assumption 3.1 ...**
>
> The strength of our approach is that it directly models the structure in the data, as embodied by assumption 3.1. This is necessary: any theory explaining why learned features have some particular structure ought to reference data structure. In reality, assumption 3.1 is more like an observation, supported explicitly in Figures 6 and 7 and implicitly throughout the paper.
>
> Clearly assumption 3.1 is never strictly exact, and only holds approximately. The theory-experiment mismatch is simply controlled by standard matrix perturbation theory. We handle this more general setting in section 4 and in appendix D.
>
> #### **it does not discuss whether such observations would be transferable or cause potential failure cases**
>
> The fundamental prediction of our theory is that if the co-occurrence of temporal markers exhibit translation symmetry, then the language model will learn Fourier representations. Whether this symmetry exists in the dataset of choice is an empirical question. For example, co-occurrence is certainly key in other domains, including computer vision: the notion of seasonality is present for images, and will lead to correlations between items entering visual scenes. As another example, prior work has shown that large text embedding models learn circular representational manifolds for hues on the color wheel, suggesting that perceptual continuums may exhibit this symmetry in very large text corpora. Extending our work to these domains is indeed very interesting, as we will indicate in the published version.

---

> > ### Author Rebuttal · Reviewer_GdUL · 2026-04-03
> >
> > I thank the authors for their response and have no further questions regarding the paper.

---

### Official Review · Reviewer_UkLF · 2026-03-13

**Soundness:** 3
**Presentation:** 3
**Significance:** 3
**Originality:** 3
**Overall Recommendation:** 4
**Confidence:** 2

**Summary:**

The authors theoretically and empirically investigate phenomenon wherein concepts encoded in textual representations exhibit surprising geometry. The authors consider cases wherein concepts can be viewed as instantiations of a latent variable whose values lie in a coordinate system fitting to the concept, eg. cyclical for months of the year. They then assume that concepts' normalized co-occurences can be accurately modeled through a kernel function applied to the concepts' distance within the chosen coordinate system. Leveraging this, it becomes possible to predict the geometry of the concepts' embeddings, with predictions seeing empirical support. That this is possible is a surprising phenomenon that seems to merit futher study.

**Compliance With Llm Reviewing Policy:**

Affirmed.

**Final Justification:**

My concerns have been addressed.

**Key Questions For Authors:**

Could the authors clarify what is meant by a semantic lattice? I think this would dramatically aid my understanding and enable a better analysis of the math at hand.

**Limitations:**

yes

**Strengths And Weaknesses:**

Soundness. I assess this as a moderate strength. Empirical results appear to support theoretical ones.

Presentation. I assess this as moderate. While sections of the paper are well-focused and one's attention need not jump between sections, there are improvements that could be made. First, it would be helpful to have somewhat more exposition on the math, eg. definitions for semantic lattices or sketches for the proofs in the main text. There is mild positioning of the work wrt concurrent work; quite simply there may be little that's meaningfully related.

Significance. I assess this as a strength. Understanding representation learning and learned representations is a pillar of the success for contemporary machine learning, and this paper opens doors to an enhanced understanding of this. Applying the techniques to representations for other domains (eg. vision) would be interesting.

Originality. This is a strength. The paper highlights novel perspectives on the geometry of learned representations.

---

> ### Author Rebuttal · Authors · 2026-03-30
>
> Thank you for finding our work significant and for helping us improve the nomenclature. We define semantic continuums at the beginning of section 3, and we implicitly define semantic lattices on line 215, left column. We will revise the main text to add explicit definitions and accessible explanations of our technical results to clarify our manuscript.
>
> A semantic lattice is a uniform lattice embedded in an abstract semantic space, where each lattice site corresponds to a single word or token. One intuitive example is a historical timeline, where years are arranged on a number line - namely, the 1D lattice of integers. Another example is the 12 calendar months, where it is natural to embed the words as 12 lattice points on a ring, based on the cyclic nature of the calendar months. Though semantic lattices are abstract mathematical objects, their geometry measurably affects the statistics of natural language: the lattice distance between words determines their relative co-occurrence probability. This intuition is formalized and generalized in Assumption 3.1 -- which is really an observation on the structure of natural data (see Figures 6 and 7). Introducing such lattices makes the connection between co-occurrence and discrete Fourier transform obvious. It is this connection that underlies our propositions 1 and 2.
>
> Using this observation, we directly predict several quantities of interest: the eigenvalue spectrum of the set of learned representation vectors (propositions 1, 2, 3), their full high-dimensional geometry (propositions 1, 2, 3, figures 1 and 2), the transferability of these vectors for downstream tasks (proposition 4, right panel of figure 2), and the robustness of our predictions to perturbations such as sampling noise and ancillary semantic signals (Section 4, appendix D, theorem 5). Such a detailed theoretical description of the learned representations has not been achieved in prior work, in any practical setting.
>
> We hope that these clarifications will satisfy your concerns and thus lead to an increased score. Otherwise, we’d be happy to answer any remaining questions you may have.

---

> > ### Author Rebuttal · Reviewer_UkLF · 2026-04-04
> >
> > Thanks for the followup. Since the math is explained either at a high level or in a way that requires significant pre-existing background to confidently understand, I maintain my confidence. Appendices B and E1 provide an ample start towards building up this mathematical landscape, and I would encourage centering their material more. Altogether, the empirical results are interesting enough that I'm not opposed to acceptance, but I'd encourage the authors to keep working on the presentation.

---

### Decision · Program_Chairs · 2026-04-30

**Decision:**

Accept (spotlight)

**Comment:**

The authors consider the symmetries in language model statistics and the emergence of geometric structure. Focusing on high-dimensional word embedding geometry, they show that translation symmetries of language statistics emerge and govern the geometry, in the case of co-occurrence of words. Moreover, they show that it is robust under perturbations of the statistics.
They provide empirical validation for their proposed framework to support the theory.

All the reviewers found the results very interesting, novel, and original. They are well motivated and well presented, and relevant and of great interest to the ICML community. I also agree with them, and I suggest acceptance.